# Biallelic *NAA60* variants with impaired N-terminal acetylation capacity cause autosomal recessive primary familial brain calcifications

Primary familial brain calcification (PFBC) is characterized by calcium deposition in the brain, causing progressive movement disorders, psychiatric symptoms, and cognitive decline. PFBC is a heterogeneous disorder currently linked to variants in six different genes, but most patients remain genetically undiagnosed. Here, we identify biallelic *NAA60* variants in ten individuals from seven families with autosomal recessive PFBC. The *NAA60* variants lead to loss-of-function with lack of protein N-terminal (Nt)-acetylation activity. We show that the phosphate importer SLC20A2 is a substrate of NAA60 in vitro. In cells, loss of NAA60 caused reduced surface levels of SLC20A2 and a reduction in extracellular phosphate uptake. This study establishes *NAA60* as a causal gene for PFBC, provides a possible biochemical explanation of its disease-causing mechanisms and underscores NAA60-mediated Nt-acetylation of transmembrane proteins as a fundamental process for healthy neurobiological functioning.

Primary familial brain calcification (PFBC), previously known as idiopathic basal ganglia calcification (IBGC) or Fahr's disease, is a genetic condition characterized by calcium deposition in the brain, including the basal ganglia, usually presenting as a combination of movement disorders, psychiatric and cognitive impairment[1,2]. Population-based genomic analysis indicates that PFBC is underestimated and underdiagnosed[3].

To date, pathogenic variants in four genes have been linked to autosomal dominant PFBC, with variants in *SLC20A2* (OMIM: 158378)[4] and *XPR1* (OMIM: 605237)[5] encoding phosphate transporters, *PDGFB* (OMIM: 190040)[6] and *PDGFRB* (OMIM: 173410)[7] encoding a growth factor and its main receptor with a critical role at the neurovascular unit. Among these, variants in *SLC20A2* account for ~45% of all genetically confirmed PFBC cases from diverse ethnicities[8]. More recently, biallelic variants in *MYORG* (OMIM: 618255)[9,10] and *JAM2* (OMIM: 606870)[11,12] have been implicated in the pathogenesis of autosomal recessive PFBC. Currently, ~50% of familial and sporadic cases remain unexplained[13] after screening of all PFBC-related genes and other mitochondrial or interferonopathies genes that can present with brain calcifications, such as *CMPK2* (OMIM: 611787)[14] or *ISG15* (OMIM: 616126)[15].

NAA60 (NatF) catalyses N-terminal (Nt)-acetylation of several transmembrane proteins with a Met-hydrophobic or Met-amphipathic-type N-terminus[16–18]. NAA60 is unique among the N-terminal acetyltransferases (NATs) for its localization to the Golgi apparatus[19], where it resides on the cytoplasmic face of the membrane[16] through peripheral binding achieved by two α-helices located at the C-terminal end[20]. Knockdown of NAA60 results in disruption of the Golgi structure and a reduced Nt-acetylation level of several membrane proteins, most typically proteins with an N-IN topology[16]. Nevertheless, no studies have established the role that this NAT and its protein-modifying capacity play in membrane protein function and human physiology.

## Results

### Biallelic *NAA60* variants lead to autosomal recessive PFBC

Aiming to identify novel pathogenic variants underlying PFBC, we first screened a cohort at UCL, comprising 78 cases from 53 families with

✉ e-mail: v.chelban@ucl.ac.uk; henriette.aksnes@uib.no; thomas.arnesen@uib.no; h.houlden@ucl.ac.uk

PFBC who were negative for pathogenic variants in genes already linked to PFBC (*SLC20A2, PDGFB, PDGFRβ, XPR1, MYORG, JAM2*). Previous extensive metabolic and mitochondrial investigations in all families excluded acquired and inherited causes of brain calcifications. We identified *NAA60* biallelic variants co-segregating with the disease in one family (F1) through a combination of whole-exome sequencing (WES) and homozygosity mapping. Then we screened for *NAA60* variants in WES data from two French brain calcification series, the pan-European SolveRD[21] project, unsolved rare disease cohorts from Turkey and Saudi Arabia. Finally, we screened the rare diseases cohort from the 100,000 Genomes Project[22] that had whole-genome sequencing (WGS) performed and data filtered as per protocol[22]. As a result of this extensive screening, we found deleterious biallelic variants in *NAA60* associated with PFBC in eight more patients from six unrelated families with autosomal recessive inheritance (F2-F7) (Fig. 1, Supplementary Fig. 1, and Table 1).

Homozygosity mapping was performed in F1-F4 (Supplementary Fig. 2). Variants were filtered by frequency (MAF < 0.001) and potential gene effects (splicing or coding, excluding non-splicing synonymous) and homozygosity coordinates, where available.

In F1, the *NAA60* c.321_327del, p.(Arg108Thrfs*3) (NM_001083601) variant segregated with the disease under a fully penetrant autosomal recessive model (Fig. 1a, b and Supplementary Fig. 1). *NAA60* c.321_327del is only present twice in the heterozygous state in gnomAD v2.1 and is absent in the homozygous state. The kinship coefficient between the siblings in F1, calculated using Peddy[23], did not exceed the expected 0.5 value, and comparing the samples in Automap[24] did not reveal a large number of shared runs of homozygosity to support close consanguinity. Although F2 and F3, recruited from France, were not known to be related, we identified the same homozygous truncating variant in *NAA60* that fully segregated with the disease in all four affected relatives: *NAA60*, c.338-1 G > C (Fig. 1a, b and Supplementary Fig. 1). No evidence of close relatedness or shared haplotype blocks surrounding *NAA60* was found in F2 and F3 (Supplementary Fig. 2), suggesting that this was a recurrent variant. In vitro RNA analysis of the F2 variant suggested that this splicing variant causes a deletion of 20 nucleotides, leading to a frameshift and premature stop codon (r.338_357del, p.(Gly113Valfs*32), Supplementary Fig. 3a–c)).

In F4, WGS performed as part of 100,000 Genomes Project, revealed the homozygous missense *NAA60* variant c.391 C > T, p.(His131Tyr) in the proband, which was heterozygous in the parents (Fig. 1a and Supplementary Fig. 1). In F5, recruited from the Solve-RD project, trio WES revealed a homozygous missense *NAA60* variant c.130 C > T, p.(Arg44Cys) in a proband with a developmental disorder, which was present in the heterozygous state in both unaffected parents (Fig. 1a and Supplementary Fig. 1). In F6, recruited from Turkey, and Family 7 recruited from Saudi Arabia, the homozygous c.50 T > G, (p.Leu17Arg) and c.428 A > C, (p.Asn143Thr) respectively were found in probands with brain calcifications and movement abnormalities (Fig. 1a and Supplementary Fig. 1). All these missense variants are absent in the homozygous state and absent or extremely rare in heterozygous state in gnomAD.

Detailed clinical descriptions can be found in Table 1, Supplementary Movie 1 and Supplementary Note 1. Six (out of 10) patients with adult-onset presented in the second and third decades of life, and the remaining four patients had onset in childhood. Those with childhood onset had associated features of global developmental delay affecting motor, learning, and language development. In the adult-onset group, the presenting features were heterogeneous, including unsteadiness (due to cerebellar ataxia and/or parkinsonism with pyramidal syndrome), seizures, and in one case (F1-II-2) acute psychosis. During the disease course, patients developed a combination of movement disorders and motor abnormalities: ataxia in 5/10 patients, parkinsonism in 4/10 patients, pyramidal syndrome in 5/10 patients, and dystonia in 2/10 patients. Half of the cases presenting

with parkinsonism were treated with levodopa, which provided only mild benefit early in the disease. Four patients exhibited psychiatric features, including psychosis, severe depression/anxiety, or ADHD, and all cases had either intellectual disability (those with childhood-onset) or developed cognitive impairment (those with adult-onset). The development of cognitive features was observed as early as five years from disease onset, and although the anterior and subcortical functions were most severely affected, the involvement was global. Mild dysmorphic features were present in half of the patients, severe migraines in three cases, seizures in three cases, and tics and stereotypies in one case.

Most *NAA60* patients showed extensive brain calcification on brain scans (Table 2) with an average total calcification score (TCS) of 51.75/80 [range 46–64] among Families 1, 2, and 3 carrying homozygous truncating variants. Symmetrical calcifications involved the basal ganglia, depth of multiple cortical sulci, supratentorial white matter, cerebellar hemispheres and vermis in all affected patients from Families 1–3 but also central pontine regions in 3 of the 7 patients with raw images available for the rating (Fig. 1b). The probands from F4, F6 and F7, carrying homozygous missense variants, exhibited smaller amounts of calcifications with bilateral lenticular calcifications (TCS of 5) at 19 years of age (in F4), bilateral hippocampal, left putamen, and periventricular white matter adjacent to the right frontal horn (TCS of 2 at age 12 years, not accounting for hippocampi, in F6) and symmetrical calcification in cerebral hemispheres, basal ganglia, depth of cortical sulci, and cerebellum (in F7). Patient from Family 7 had a CT angiography of the brain at the age 27 years old, which showed multifocal narrowing of the distal intracranial arteries with likely secondary intravascular calcium deposition in the absence of cardiovascular risk factors (Fig. 1b (VIIb)). No conclusive brain calcification imaging was present in the patient from F5 at the age of 12 years. Although calcifications are present in most adults carrying PFBC-causing variants[2] the timing of when the first neuroimaging signs of calcifications can be identified is not clear. Despite extensive genetic analysis in this case, we cannot exclude any additional, yet unidentified genetic variants, potentially contributing to some of the magnetic resonance imaging (MRI) features in this case that were absent in the other cases reported here (microcephaly and polymicrogyria).

The *in-silico* prediction of all reported *NAA60* variants is in Supplementary Table 1. *NAA60* variants of Families 1, 2, and 3 cause large truncations in the NAA60 protein (Fig. 2a–c). These truncations omit half of the GNAT domain, which forms a recognizable tertiary structure common to all N-acetyltransferases, including NATs[25], and are likely to severely affect the binding of both the substrate N-terminus and the acetyl donor acetyl-CoA. For the *NAA60* missense variants p.His131Tyr (F4), p.Arg44Cys (F5), p.Leu17Arg (F6), and p.Asn143Thr (F7), sequence alignment of various species showed conservation of His131, Leu17, Asn143 and a somewhat conserved residue of Arg44 (Fig. 2a). Moreover, DynaMut predictions indicated that all reported missense variants have a similar degree of increased intramolecular stability and decreased molecule flexibility (Supplementary Fig. 4).

Our structural predictions showed that the sites affected in F4 and F5 are positioned on the NAA60 protein surface and do not directly contact Ac-CoA or the peptide substrate (Fig. 2d). However, in contrast to WT, *NAA60* His131Tyr (F4) does not exhibit hydrogen binding to Ala128 but is rather predicted to interact with Asp92; *NAA60* Arg44Cys (F5) shows the same interaction residues as WT, with an additional binding to Asp40, thus strengthening the intramolecular interaction and decreasing the flexibility of this protein variant (Fig. 2e). For the variant from F6, DynaMut prediction showed the Leu17Arg substitution remarkably affects several interacting forces with adjacent residues. While Leu17 shows diverse hydrophobic contacts with Leu55, Gly65, Phe53, and ionic and carbonyl contacts with Ser54, the Arg17 substitution only demonstrates an interacting force with Ser54 (Fig. 2e). The residue affected in F7, Asp143 is

conserved in all presented orthologs (Fig. 2a). This is an area shown to be important for substrate-specific acetylation[26]. Due to large structural differences compared to Asn143, the Thr143 substitution in NAA60 F7 was predicted to have reduced interaction with several

residues (Fig. 2e), including lost interactions with Thr142, Val139, and Thr145 as a result of the substitution, while interactions to Leu140, important for substrate recognition and homodimerization[26], remained intact.

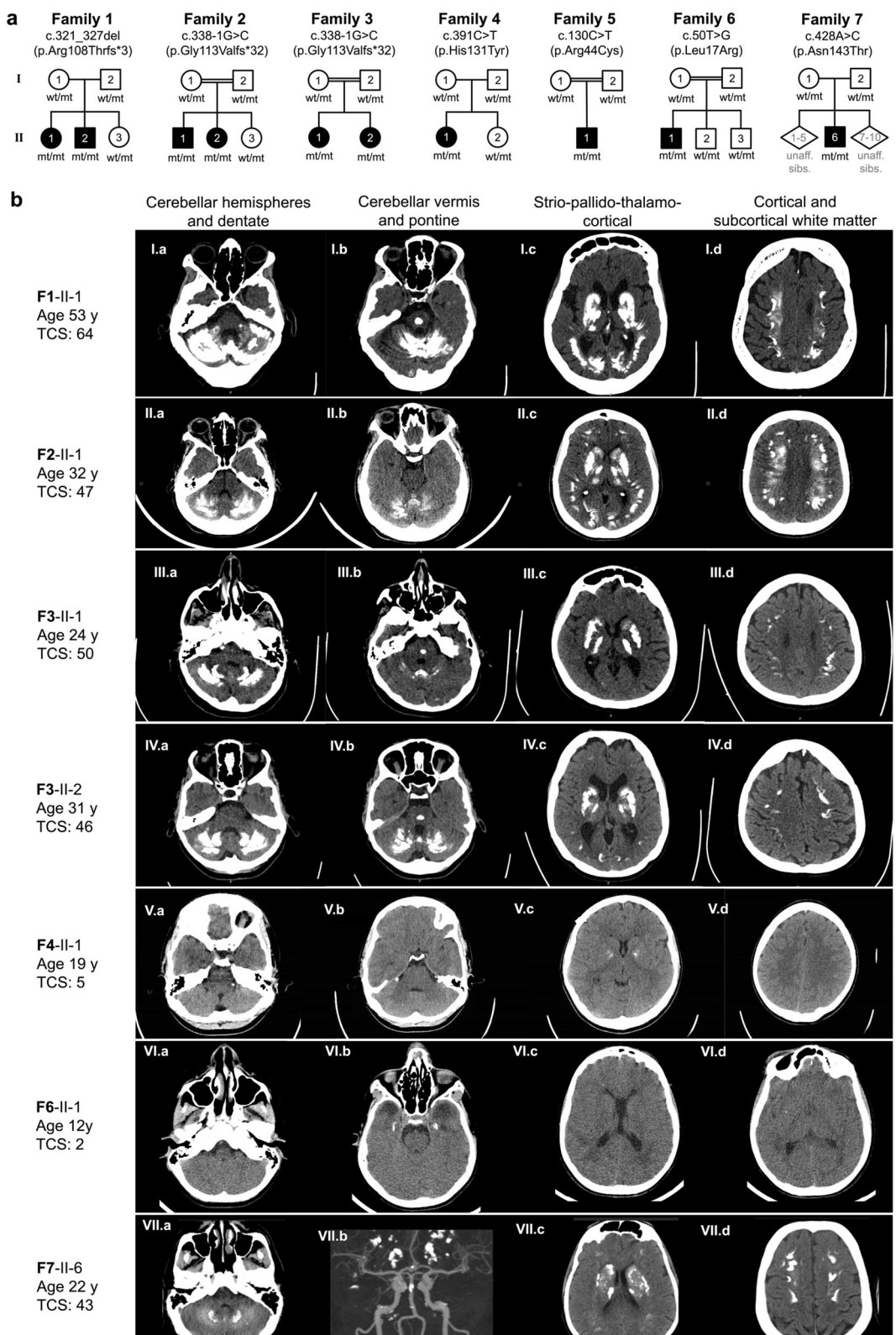

**Fig. 1 | *NAA60* biallelic variants lead to primary familial brain calcifications.**
**a** Pedigrees of the families with *NAA60* biallelic variants identified in this study. **b** CT scan axial views illustrating brain calcification in patients with *NAA60* biallelic variants. Image VII.b shows CT angiography of the brain revealing multifocal narrowing of the distal intracranial arteries, compatible with an intracranial vasculopathy likely secondary to intravascular calcium deposition. TCS total calcification score.

**Table 1 | Main clinical findings in *NAA60*-related disease**

| Family | 1 | 1 | 2 | 2 | 3 | 3 | 4 | 5 | 6 | 7 |
|---|---|---|---|---|---|---|---|---|---|---|
| Individual | F1-II-1 | F1-II-2 | F2-II-1 | F2-II-2 | F3-II-1 | F3-II-2 | F4-II-1 | F5-II-1 | F6-II-6 | F7-II-6 |
| Origin of case | UK | UK | France/Algeria | France/Algeria | France/Morocco | France/Morocco | UK/India (Gujarati) | Turkey | Turkey | Saudi Arabia |
| Consanguinity | No | No | Yes | Yes | Yes | Yes | No | Yes | Yes | Yes |
| cDNA sequence | c.321_327del | c.321_327del | c.338-1 G>C | c.338-1G>C | c.338-1G>C | c.338-1G>C | c.391 C>T | c.130 C>T | c.50 T>G | c.428 A>C |
| Amino-acid change | (p.Arg108Thrfs*3) | (p.Arg108Thrfs*3) | p.(Gly113Valfs*32) | p.(Gly113Valfs*32) | p.(Gly113Valfs*32) | p.(Gly113Valfs*32) | (p.His131Tyr) | (p.Arg44Cys) | (p.Leu17Arg) | (p.Asn143Thr) |
| Zygosity | Homozygous | Homozygous | Homozygous | Homozygous | Homozygous | Homozygous | Homozygous | Homozygous | Homozygous | Homozygous |
| Age of onset | Early twenties | Early twenties | Early thirties | Developmental delay from birth. Onset motor symptoms at age 20 | Mild intellectual disability from birth. Onset of motor symptoms at age 31 | Mild intellectual disability from birth. Onset of motor symptoms at age 24 | Mild symptoms at birth. Onset of motor symptoms at age 10 | Developmental delay from birth | Developmental delay from birth. | Late twenties |
| Motor features and movement disorders | Extrapyramidal and cerebellar syndrome | Extrapyramidal and cerebellar syndrome | Extrapyramidal and cerebellar syndrome, dystonia | No | Pyramidal, extra-pyramidal and cerebellar syndrome | Pyramidal and cerebellar syndrome | Hypotonia, tics and stereotypies, tongue dyskinesia, dystonia, chorea, and pyramidal syndrome | Pyramidal syndrome, dystonia | Pyramidal syndrome | Pyramidal and cerebellar syndrome. |
| Psychiatric features | No | Psychosis. schizophrenia | No | No | Mild intellectual disability, depression, anxiety | Mild intellectual disability | ADHD | No | Mild intellectual disability, ADHD | No |
| Cognitive features | Cognitive impairment | Cognitive impairment | Learning difficulties | Mild frontal syndrome | Mild learning difficulties | Mild learning difficulties | Learning difficulties | Delayed language development | Mild learning difficulties | Mild cognitive impairment |
| Disease duration at last examination | 17 years | 15 years | 47 years | 6 years | 39 years | 33 years | 6 years | 9 years | 12 years | 2 months |
| Findings at the last examination | Severe cerebellar syndrome and parkinsonism, anarthria, unable to swallow | Anarthria, severe parkinsonism, unable to swallow | Cerebellar syndrome, parkinsonism, dystonia | Unremarkable | Pyramidal and cerebellar syndrome, Mild intellectual disability, depression, anxiety | Pyramidal syndrome, mild parkinsonism, moderate cerebellar ataxia | NA | Pyramidal syndrome, delayed motor milestones, dystonia, quadriparesis | Pyramidal syndrome | Pyramidal and cerebellar syndrome |

**Table 2 | Neuroimaging and additional clinical findings in *NAA60*-related disease**

| Individual | F1-II-1 | F1-II-2 | F2-II-2 | F2-II-1 | F3-II-1 | F3-II-2 | F4-II-1 | F5-II-1 | F6-II-6 | F7-II-6 |
|---|---|---|---|---|---|---|---|---|---|---|
| cDNA sequence | c.321_327del | c.321_327del | c.338-1 G > C | c.338-1 G > C | c.338-1 G > C | c.338-1 G > C | c.391 C > T | c.130 C > T | c.50 T > G | c.428 A > C |
| Amino acid change | (p.Arg108Thrfs*3) | (p.Arg108Thrfs*3) | p.(Gly113Valfs*32) | p.(Gly113Valfs*32) | p.(Gly113Valfs*32) | p.(Gly113Valfs*32) | (p.His131Tyr) | (p.Arg44Cys) | (p.Leu17Arg) | (p.Asn143Thr) |
| Zygosity | Homozygous | Homozygous | Homozygous | Homozygous | Homozygous | Homozygous | Homozygous | Homozygous | Homozygous | Homozygous |
| Age at CT scan | 64 years | NA | NA | 32 years | 24 years | 31 years | 19 years | 12 years | 12 years | 43 years |
| Areas of calcifications on CT scans | Extensive bilateral basal ganglia, pons, cerebellum | Extensive bilateral basal ganglia, pons, and cerebellum | Extensive bilateral basal ganglia, subcortical white matter | Extensive bilateral basal ganglia, cerebellum, cortical regions, subcortical white matter | Extensive bilateral basal ganglia, pons, cerebellum cortical regions | Extensive bilateral basal ganglia, pons, cerebellum cortical regions | Globus pallidus bilaterally | NA | Bilateral mesial temporal lobes, left putamen, and periventricular white matter adjacent to the right frontal horn | Extensive bilateral basal ganglia, cerebellum. |
| TCS | TCS = 64 | NA | NA | TCS = 47 | TCS = 52 | TCS = 46 | TCS = 5 | NA | TCS = 2 | TCS = 43 |
| MRI features in addition to calcification | Central volume loss. Grossly thickened skull. | NA | None | NA | NA | NA | Supratentorial hyperintensity in the frontoparietal subcortical white matter bilaterally | Polymicrogyria | Hyperintensity in the bilateral parietal and posterior temporal lobes. Gliosis in the bilateral temporal poles. Bilateral cerebral white matter volume loss most notable in the periventricular regions of the parietal and occipital lobes | Multifocal, chronic and subacute ischemic changes with intracranial vasculopathy |
| Additional clinical features | Migraine. Required PEG | Required PEG | Mild dysmorphic features (macrocrania with an oblong face), seizures | Mild dysmorphic features (macrocrania with an oblong face). Seizures | Facial dysmorphia, genu valgum, varus of feet, aortic coarctation, migraine, seizures | Macrocrania, oblong face, agenesis of 2nd, 3rd and 4th right toes, hypoplasia of 2nd, 3rd and 4tg left toes, hypopigmented regions on both upper limbs, migraine | Glue ears, hyperacusis, short stature, dysmorphic features (low set ears, almond shaped eyes), congenital cataracts | Proximal upper limb amyotrophy, dysmorphic features (microcephaly, exophthalmos), seizures | Microcephaly, high palate, strabismus, seizures | No |

*NA* not available, *ADHD* attention deficit hyperactivity disorder, *TCS* total calcification score, *PEG* percutaneous endoscopic gastrostomy.

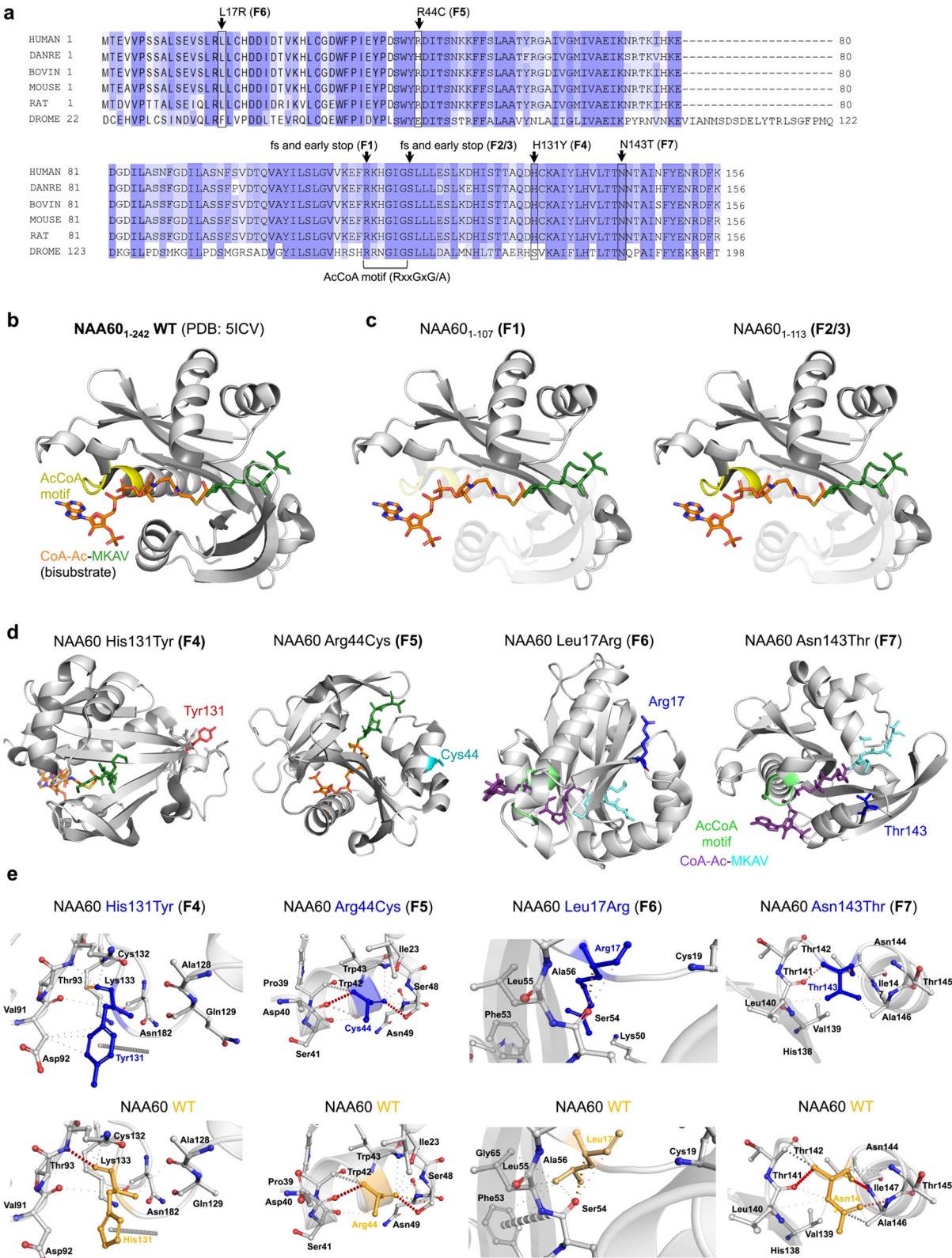

**Fig. 2 | NAA60 protein sequence and structural features predict functional implications of *NAA60* variants. a** Alignment of the *NAA60* amino acids corresponding to the GNAT fold across different species shows strong conservation of the regions affected by *NAA60* variants found in PFBC cases of the seven families (Family 1–7). UNIPROT accession numbers: human, Q9H7X0; *D. rerio* (DANRE), A3KPA3; bovine (BOVIN), Q17QK9; mouse, Q9DBU2; rat, Q3MC1; *D. melanogaster* (DROME), Q95SX8. **b** Structure of wild-type NAA60 (PDB: 5ICV, amino acids 4–184 shown in gray from three different angles) with the bound CoA-Ac-MKAV bisubstrate analog in stick representation. **c** PDB structure modified in PyMol to visualize the truncated proteins predicted from the PFBC *NAA60* variants F1 and F2/3. The Ac-CoA motif is indicated in yellow. **d** PDB structure modified in PyMol to visualize the PFBC *NAA60* missense variants in F4-F7. **e** Interaction figures derived from DynaMut predictions displaying interaction forces between the mutated residues in F4-F7 and surrounding residues.

### *NAA60* variants lead to loss-of-function with lack of protein N-terminal-acetylation activity

The gene products of all *NAA60* variants reported are shown in Fig. 3a. To assess whether the *NAA60* variants alter the subcellular distribution and function of this N-alpha-acetyltransferase, we expressed constructs with sequences corresponding to those in affected cases. FLAG-NAA60-F1 was distributed in the cytosol and nucleus (Fig. 2b), like in previously investigated truncations[20]. Similar results were obtained for the other truncated variants (F2/F3), whereas the missense variants (F4, F5, F6, and F7) exhibited intact subcellular localization (Fig. 3b).

From the experiments in Fig. 3b, it became apparent that there were differences in the degree of expression among the *NAA60* variants, with some producing a very low amount of protein, even from a CMV promoter. We hypothesized that this was due to degradation and therefore used the proteasomal inhibitor MG132. Indeed, the variants found in individuals from Families 1, 2, and 3, but not 4 and 5, were stabilized by the proteasomal inhibitor MG132 (Fig. 3c), indicating that these undergo rapid proteasomal degradation in cells.

For the families with available primary cells isolated, we assessed the endogenous NAA60 level using an NAA60 antibody. Primary lymphoblasts from the case in F7 and primary dermal fibroblasts from the two F1 cases and their heterozygous mother were tested alongside matched healthy controls. Noteworthy, the missense mutation Asn143Thr in F7 entailed a dramatically reduced, possibly absent, NAA60 protein (Fig. 3d).

Next, we assessed whether the NAA60 protein variants have the capacity to Nt-acetylate classic NAA60 substrate N-termini. When immunoprecipitated *NAA60* variants were incubated with MAPL (Met-Ala-Pro-Leu-) substrate peptide and acetylation buffer mix, NAA60-F1 was completely unable to perform Nt-acetylation of this classical NAA60-type substrate (Fig. 3e). Similarly, albeit less striking, results were obtained for NAA60-F2/3, which appeared to demonstrate some biochemical activity in this pure in vitro assay. Nevertheless, given the cellular instability (Fig. 3c, d), cellular mislocalization (Fig. 3b), and the major GNAT fold truncations (Figs. 2 and 3a), these severely truncated proteins are unlikely to retain any NAT activity in vivo. The NAA60-F4 and -F5 also demonstrated diminished Nt-acetylation capacity in this assay (Fig. 3e). Overall, these results are consistent with the reduced N-terminal acetylation of NAA60 substrates as a cause of PFBC.

The microscopy in Fig. 3b indicated that NAA60 did not perfectly co-localize with the Golgi marker GM130. Therefore, we performed structured illumination microscopy and co-localization analysis, which revealed that FLAG-NAA60-WT was co-distributed with the cis-Golgi marker GM130, as it localized to structures closely associated to, but not completely overlapping with the structures stained by GM130, (Fig. 4a, and Supplementary Movie 2). This likely means that NAA60 prefers another sub-compartment of the Golgi. Nonetheless, we found that GM130 is a valid marker for investigating intact versus disturbed Golgi localization of NAA60 and for assessing Golgi morphology. Previously, using GM130 as a marker, it was shown that knockdown of *NAA60* in HeLa cells results in a disruption of the Golgi ribbon architecture[16]. To determine whether the *NAA60* frameshift variants reproduce this phenotype, we performed immunofluorescence microscopy of primary dermal fibroblasts from *NAA60*−/− cases from F1. However, the Golgi structure appeared intact in all cells from both affected and unaffected individuals (Fig. 4b). We reasoned that this discrepancy with previous findings could be due to cell type differences or that it could be explained by differential effects of knockdown vs knockout. We, therefore, investigated a CRISPR/Cas9-generated *NAA60* knockout cell line harboring a frameshift variant at a position close to that of NAA60-PFBC in Families 1, 2 and 3 (Fig. 4c). Of note, these HAP1 *NAA60*−/− cells also showed an intact Golgi apparatus (Fig. 4d) where *NAA80* knockout was used as a positive control for Golgi fragmentation in the HAP1 cell line[27]. We next extended this investigation by performing the same shRNA knockdown experiment as previously done in HeLa cells, but

now in dermal fibroblasts from healthy controls. Here, we observed that *NAA60*-shRNA-transfected dermal fibroblast controls displayed fragmented Golgi (Fig. 4e), similar to the previous work in HeLa cells[16]. Thus, we found support for the idea that Golgi fragmentation occurs in RNA silencing experiments but not in cells with permanent NAA60 absence due to an early frameshift in the genomic DNA.

## Loss of NAA60 function likely cause PFBC through a P$_i$ homeostasis pathway

NAA60 likely has many substrates[16] and all membrane proteins associated with PFBC may be considered potential NAA60 substrates: SLC20A2, XPR1, MYORG, JAM2, and PDGFRβ. Therefore, we tested the ability of NAA60 to Nt-acetylate peptides corresponding to the N-termini of these proteins in vitro (Fig. 5a). Among them, SLC20A2 was the best NAA60 substrate (Fig. 5b), while significant Nt-acetylation was also observed for the MYORG N-terminus. The N-terminus of SLC20A2 is a MAMDE. Since it is possible that this may exist in the iMet-processed variant, two versions of this peptide were tested. The peptide MAMD was again Nt-acetylated to the same degree as the classical NAA60 substrate MAPL, previously found to be acetylated by NAA60 both in cells and in vitro[18]. In contrast, the iMet-processed variant, AMDE, was not a good in vitro substrate of NAA60, performing similarly to the negative controls normally Nt-acetylated by other NATs in cells (Fig. 5c). Our data indicate that SLC20A2 (MAMD N-terminus) is a potential in vivo substrate of NAA60.

As SLC20A2 was found to be a possible substrate of NAA60, we investigated whether an impaired *NAA60* variant affects the presence of SLC20A2 at the dedicated location at the plasma membrane (PM). We isolated PM proteins with surface biotinylation on primary patient fibroblasts obtained from homozygous F1 (Fig. 6a). This revealed a significantly decreased surface level of SLC20A2 in *NAA60*−/− cells from F1-II-1 and F1-II-2, while cells from a heterozygous carrier had similar surface levels to the unrelated control (Fig. 6b, c).

Next, we investigated whether the reduced surface level of SLC20A2 in cells from *NAA60*-related PFBC cases may be associated with defects in P$_i$ transport across the cell membrane. We measured the concentration of extracellular P$_i$ in the cell culture medium similar to in previous studies[13] (Fig. 6d). Medium from *NAA60* F1 cases contained significantly higher P$_i$ levels than medium from age-matched controls and an *SLC20A2* missense mutant case associated with impaired function (Fig. 6e). We assessed the cell capacity for P$_i$ depletion and found that there was a lower expenditure of P$_i$ in cell cultures from *NAA60* mutant cases (7.9%) compared to *SLC20A2* mutant cases (36.1%) and controls (43.3%) (Fig. 6e'). In addition, investigation of fresh peripheral blood mononuclear cells from an affected patient in F2 confirmed a similar effect on cellular P$_i$ homeostasis (Fig. 6f, f').

## *NAA60* and *SLC20A2* co-expression networks

Given NAA60's effect on the SLC20A2 surface level and the similar cellular phenotypes of *NAA60* and *SLC20A2* mutations, we next looked at the tissue distribution profiles of these two genes According to GTEx, both *NAA60* and *SLC20A* are expressed in all tested human tissues, including brain (Figure S5a). We then created co-expression networks that include *NAA60* and *SLC20A2*. This resulted in five co-expression modules containing *SLC20A2* across the five brain regions and three modules containing *NAA60*. This revealed that within three of the regions (cerebellum, nucleus accumbens, and caudate), *NAA60* and *SLC20A2* genes are found within different modules in each tissue-specific network. However, for the nucleus accumbens network, these two modules are very close (Supplementary Fig. 5d–f). The grey60 module, containing *SLC20A2* module was of particular interest as it was significantly enriched for genes associated with human phenotype (HP) terms like basal ganglia calcification, HP:0002135, ($P < 3.34 \times 10^{-4}$) and vascular system, with terms like abnormal vascular morphology

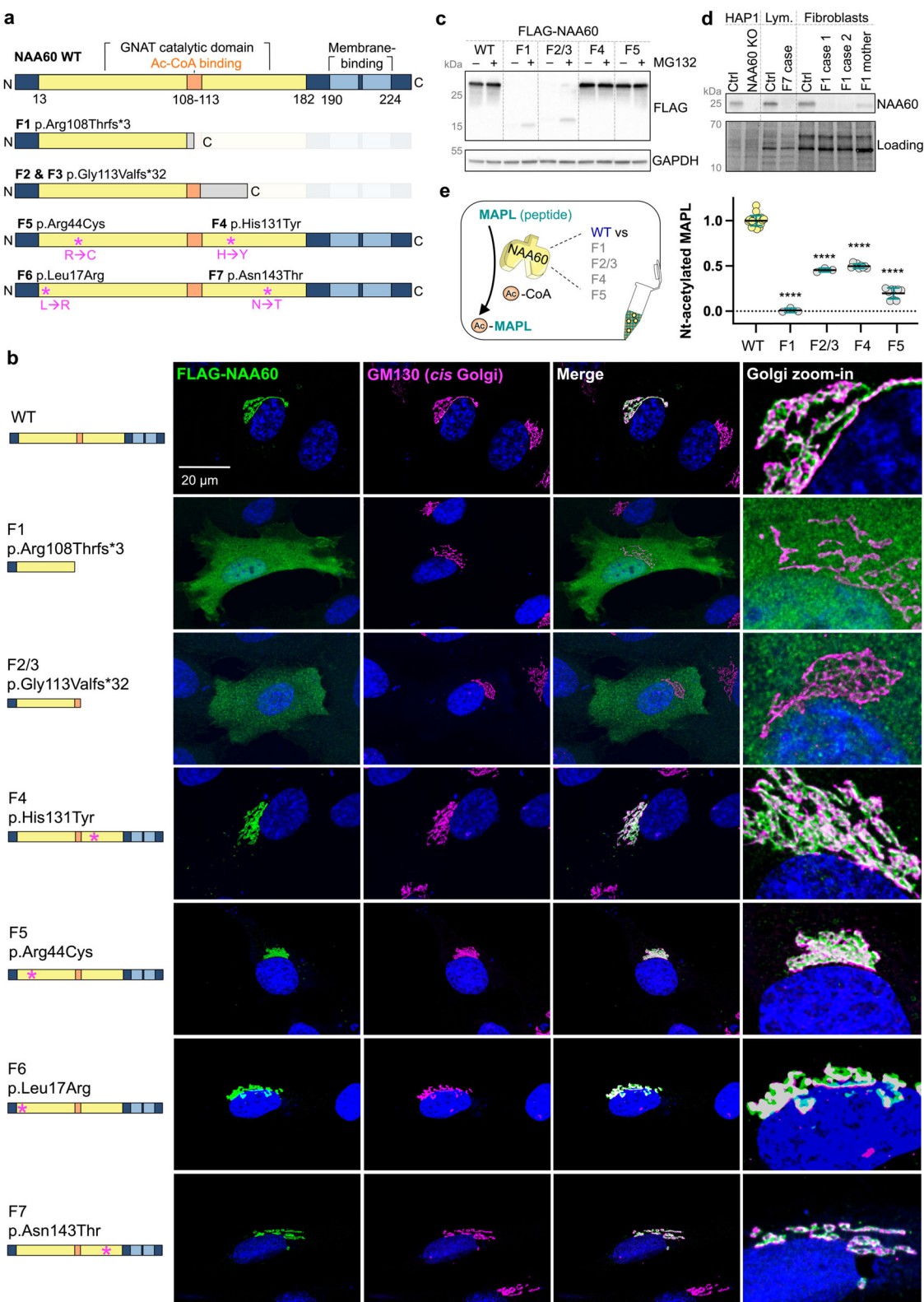

(P < 2.97 × 10⁻⁵) HP:0025015, and abnormal cerebral vascular morphology, HP:0100659 (P < 2.656E-4). Specific cell marker enrichment analysis of the same module shows enrichment for endothelial (p < 2.945E-44), mural (p < 2.440E-16), and microglial cell markers (p < 8.023E-7). The blue module, which contains the *NAA60*, is enriched with genes associated with neurodevelopmental delay HP:0012758, (p < 8.90E-12), morphological central nervous system abnormality, HP:0002011 (p < 1.45E-09), abnormality of movement,

HP:0100022 (p < 6.80E-07), and abnormal muscle tone, HP:0003808 (p < 4.64E-13). This module is clearly neuronal as it is exclusively and highly enriched with these cell markers (p < 5.72E-37) (Data File 1). Overall, these findings suggest that while the expression pathways of *SLC20A2* and *NAA60* are distinct, they are closely related, especially in the nucleus accumbens. Furthermore, this data places *NAA60* within a network of genes involved in morphological brain structure and neurodevelopment in humans, particularly within the neurons from the

**Fig. 3 | Mutated NAA60 proteins fail to associate with the Golgi, are highly unstable and lack intrinsic enzymatic activity. a** Predicted proteins resulting from the PFBC variants in *NAA60* described here. WT NAA60 contains 242 amino acid (aa) residues and is part of the N-terminal acetyltransferase (NAT) family. These proteins share a distinct secondary structural GNAT domain (aa 13–182 of NAA60, indicated in yellow), comprising a critical binding site for the acetyl-CoA donor (aa 108–113 of NAA60, indicated in orange). In addition, NAA60 has a C-terminal extension unlike other NATs in which two Golgi membrane-binding amphipathic helices are found (aa 190–224, indicated in light blue). **b** FLAG-NAA60 wild-type and FLAG-NAA60-PFBC patient variants were expressed in RPE-1 cells, immunostained, and imaged with a Leica SP8 confocal microscope using the Lightning module for optimized resolution. The scale bar is 20 μm for all except for the magnified frames. Images are representative >100 transfected cells. **c** FLAG-NAA60 wild-type and FLAG-NAA60-PFBC variants were expressed in HEK293FT cells, and six hours prior to harvesting, cells were treated with 5 μM proteasomal inhibitor MG132. Result shown is representative of at least three independent experiments. **d** The indicated cells were lysed and endogenous NAA60 levels were assessed by western blot. HAP1 Ctrl and *NAA60* KO cells were used as control for the antibody. Lym. = primary lymphoblasts. Fibroblasts = primary dermal fibroblasts. Representative of three experiments. **e** NAA60 variant proteins lack intrinsic enzymatic activity. WT and variant constructs were expressed in HEK293T cells and immunoprecipitated (IP) to produce the proteins for an enzyme activity test as illustrated. NAA60 IP products were tested for their ability to Nt-acetylate the NAA60-type substrate peptide MAPL in a classical in vitro [14 C] Nt-acetylation assay. For each condition, at least three independent replica experiments were included ($n = 3$ for F1 and F2/3, $n = 9$ for F4 and F5), pretests were performed to be able to apply equal enzyme inputs, and a final normalization to enzyme input was performed. **** indicates $p < 0.0001$ by impaired two-tailed t-test with Welch's correction testing each mutant against the WT. Source data are provided in the Source Data file. Parts of the figure were drawn by using elements modified with text, markings, and annotations from Servier Medical Art, provided by Servier, licensed under a Creative Commons Attribution 3.0 (https://creativecommons.org/licenses/by/3.0/).

caudate and nucleus accumbens of the basal ganglia (Fig. 6g, Supplementary Fig. 5), in keeping with the human *NAA60*-related phenotype.

## Discussion

In this study, we show that biallelic variants in *NAA60* lead to recessive PFBC in ten individuals from seven unrelated families. The patients with *NAA60*-related brain calcification displayed their first clinical signs starting from early childhood to early twenties. We report a variable combination of clinical features, including ataxia, parkinsonism, headache, psychiatric and cognitive deficits with a high intrafamilial phenotypic variability and age at onset. However, as the disease progresses, the phenotype becomes more uniform, with an association of cerebellar, pyramidal, and parkinsonian syndromes with poor response to levodopa, consistent with patients with variants in other PFBC-related genes, especially other autosomal recessive forms[1]. Of note, the missense variants, F4 and to some degree F5, appeared to result in less severely reduced NAA60 protein function in vitro than truncating, likely complete loss-of-function, high-unstable variants in F1 and F2/3. Interestingly, the proband from F4 exhibited only limited calcifications of both lenticular nuclei at age 19 years. Although this calcification pattern is clearly abnormal at this age[28], it was less severe in intensity and in extension than F1 and F2/3, suggesting a putative genetic-radiological correlation. The proband in F5 did not show any calcifications on CT at the age of 12 years. The clinical presentation appeared to be more complex than the other patients that we ascertained. Although the complex clinical presentation raises the possibility of a co-existing diagnosis, exome sequencing did not reveal any additional variant that would explain the other phenotypic features. However, as our data strongly suggests that this *NAA60* variant is leading to abnormal NAA60 enzymatic activity combined with the young age at which the CT scan was performed, we have included this variant here, and follow-up of this patient will be important to conclude the presence of abnormal brain calcification during aging. Interestingly, patient from Family 6 exhibited very small calcifications at the age of 12, including two calcified spots in the periventricular white matter and in the left putamen, next to the external capsule. Such a pattern is abnormal in patients of this age. Imaging of both F5 and F6 patients should be followed up to determine whether they will develop typical basal ganglia calcification and even more extensive later or not. The CT scans reported here are some of the youngest FPBC scans reported in the literature. Therefore, young cases from F5 and F6 should be followed up to determine the pattern of progression of calcification on neuroimaging over time.

Gene expression data (GTEx and BRAINEAC, Supplementary Fig. 5) suggest that all brain regions express *NAA60*, with the highest expression in the putamen, followed by the hippocampus, cerebellum, occipital cortex, and thalamus. Notably, these areas are mirrored in the clinical phenotype and distribution of calcification on neuroimaging assessment in *NAA60*-related cases. Interestingly, the central pontine calcification identified in 3 of the 7 cases with available imaging was previously reported only in autosomal recessive *MYORG*- and *JAM2*-related PFBC[9,29] suggesting a regional vulnerability of the pons to this disease pathway. Furthermore, based on *in-silico* analysis and gene networks and co-expression data in humans, we show that although *NAA60* and *SLC20A* belong to different expression pathways, they work closely in a network of genes already linked with epithelial-mesenchymal transition into osteosarcoma (*ARMC8,OMIM:618521*), calcium homeostasis and Ca²⁺ signaling (*BAG5,OMIM:619747; RTN3, OMIM:604249*), cargo-recognition complex critical for endosome-trans-Golgi trafficking (*VPS35, OMIM:601501*) and protein ubiquitination and degradation (*CUL1, OMIM:603134*), which reflect the biological pathways implicated in our *NAA60* analysis. Interestingly, the *NAA60-SLC20A2* network from the caudate tissue is already linked to genes implicated in Mendelian disorders associated with clinical overlap with the *NAA60*-related phenotype, such as developmental delay, intellectual disability, variable dysmorphic facial features (*NARS,OMIM:108410; USP7,OMIM:616863; CSNK2A1, OMIM:617062*), and parkinsonism (*VPS35*) (Fig. 6g and Supplementary Fig. 5g). This data suggests that within this network there may be other genes involved in neuronal maintenance that could cause a similar disease phenotype to be discovered. It, however remains to be firmly established whether SLC20A2 may be a direct substrate of NAA60-mediated N-terminal acetylation or whether their functional association is indirect or perhaps independent of Nt-acetylation. Since NATs do not typically co-precipitate with their substrates, combined with the limited performance of currently available antibodies, any future studies investigating SLC20A2 as a potential substrate of NAA60 is best done using differential Nt-proteomics on cells expressing or lacking NAA60.

NAA60's physiological connection with PFBC and the development of calcium deposits likely occurs through its ability to Nt-acetylate membrane proteins. Prior analysis of NAA60-depleted human A-431 cells revealed 23 substrates of NAA60. All identified substrates were transmembrane proteins, and further membrane proteins were predicted as substrates based on the NAA60 preferences revealed[16]. SLC20A2 has the strongest physiological connection to the development of brain calcifications. Recent in vitro studies have suggested a role for this solute carrier in maintaining phosphate homeostasis in cerebrospinal fluid, indicating that *SLC20A2* variants severely impaired phosphate transport in the brain[30]. The role of abnormal brain phosphate homeostasis in PFBC is further supported by pathological studies in other PFBC cases due to variants in genes such as *MYORG*, showing that the intracranial precipitates contain calcium phosphate[9]. Here, we showed that the SLC20A2 N-terminus is acetylated by NAA60 in vitro. Moreover, in cells isolated from *NAA60*

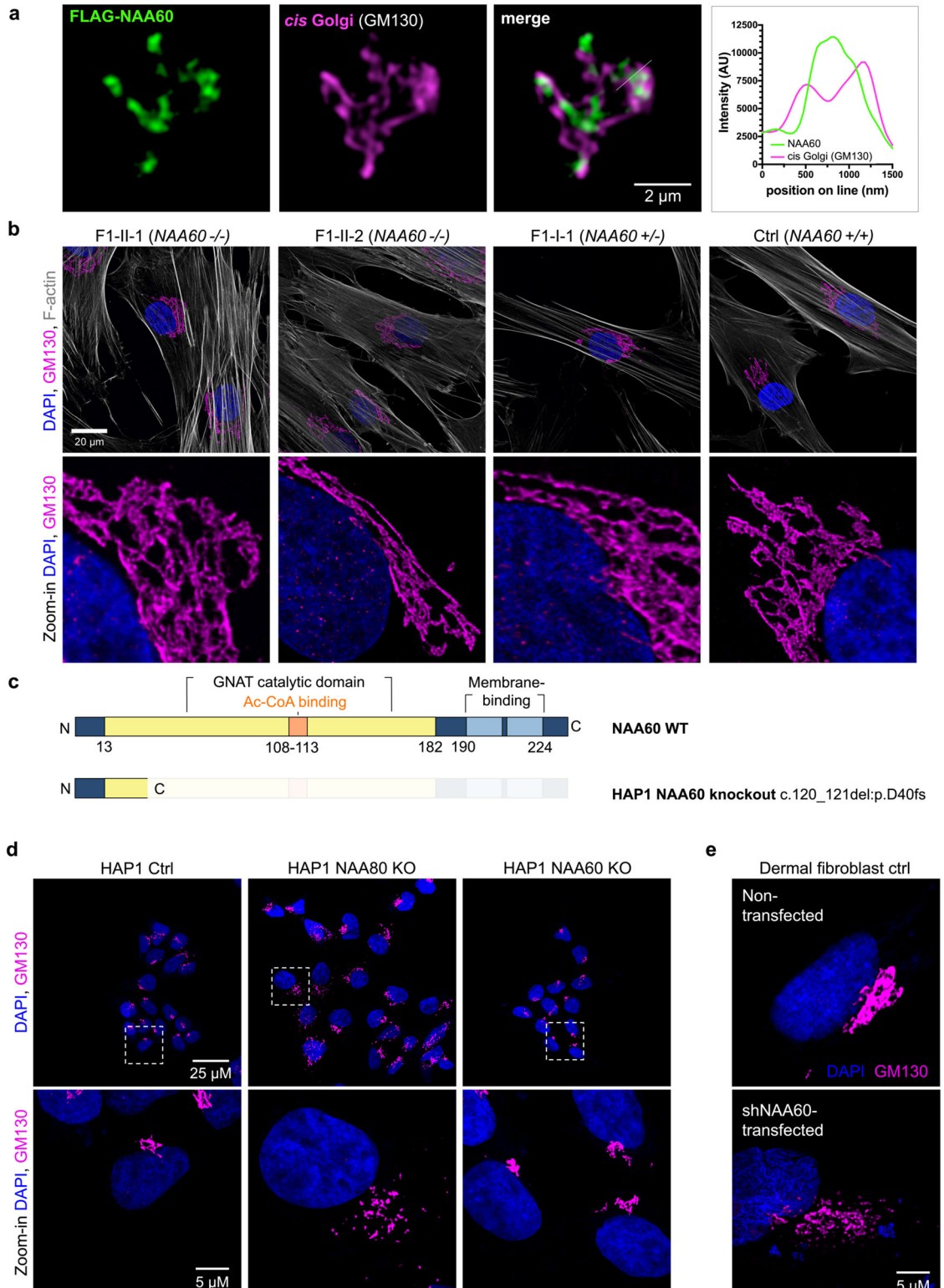

cases, surface levels of SLC20A2 were reduced along with a reduction in the ability to deplete $P_i$ from culture medium. These findings suggest that SLC20A2 subcellular targeting/function on the PM requires NAA60-catalyzed Nt-acetylation (Fig. 7). Recently, an interplay between SLC20A2 and XPR1 was unveiled, with functional transport activities that appeared to control cellular phosphate homeostasis, although only defects in *XPR1* led to increased global phosphate concentrations, which is in contrast to defective PFBC-*SLC20A2* variants, which affected neither phosphate and ATP levels nor phosphate

efflux[31]. This is consistent with our results, with variants in *NAA60* affecting SLC20A2 PM localization and cellular $P_i$ homeostasis. Moreover, a complex PFBC genotype-phenotype spectrum emerges as cases that involve dose-dependent variants have been associated with a severe clinical and neuroimaging spectrum[32]. *SLC20A2* heterozygous mutations are associated with the adult-onset phenotype, while homozygous *SLC20A2* variants resulted in severe phenotype including growth retardation, microcephaly, and convulsion[33], similar to some of the *NAA60* variants reported here.

**Fig. 4 | *NAA60* knockdown but not *NAA60* knockout cells have Golgi fragmentation. a** NAA60-FLAG-positive compartments localized close to compartments stained by the cis-Golgi marker GM130 but did not perfectly colocalize. HeLa cells were transfected with FLAG-NAA60, fixed, and stained with anti-FLAG and anti-GM130 followed by imaging using an OMX 3D-SIM microscope. Images are shown as z-stack max intensity projection and are representative of <100 cells inspected. **b** Golgi fragmentation was not observed in fibroblast cells derived from Family 1. Cells were fixed and stained for the cis-Golgi marker GM130 and with DAPI for visualization of the nucleus and phalloidin for F-actin. Imaging was performed with a Leica SP8 using the Lightning module for deconvolution and optimized resolution. Representative of at least three independent experiments. The scale bar is 20 μm for all except for the magnified frames. **c** Schematic overview of the predicted protein resulting from the frameshift introduced in HAP1 *NAA60* KO cells. Compare to the PFBC variants shown in Fig. 2a. **d** Golgi fragmentation was not

observed in HAP1 *NAA60* knockout cells. Ploidy-controlled HAP1 CTRL, *NAA60* knockout, and *NAA80* knockout cells were fixed and stained for the cis-Golgi marker GM130 and with DAPI for visualization of the nucleus. *NAA80* knockout was used as a positive control for Golgi fragmentation in the HAP1 cell line. The scale bar is 25 μm for all images on top and 5 μm for all images at the bottom. **e** Dermal fibroblast healthy control cells were transfected with shNAA60+RFP plasmid 72 h prior to fixation. Images were acquired with a Leica TCS SP8 confocal microscope at 100x using a zoom factor of 0.75 and a z-step size of 0.25 μm. The scale bar is 5 μm for both images shown. The degree of Golgi intactness was evaluated by microscopy using a Leica DMI6000 B wide-field fluorescence microscope with an HCX PL APO 100 × 1.4 NA oil objective. Images are representative of at least 100 inspected cells (HAP1) and at least 30 transfection-positive dermal fibroblast cells where near all transfected cells had the phenotype.

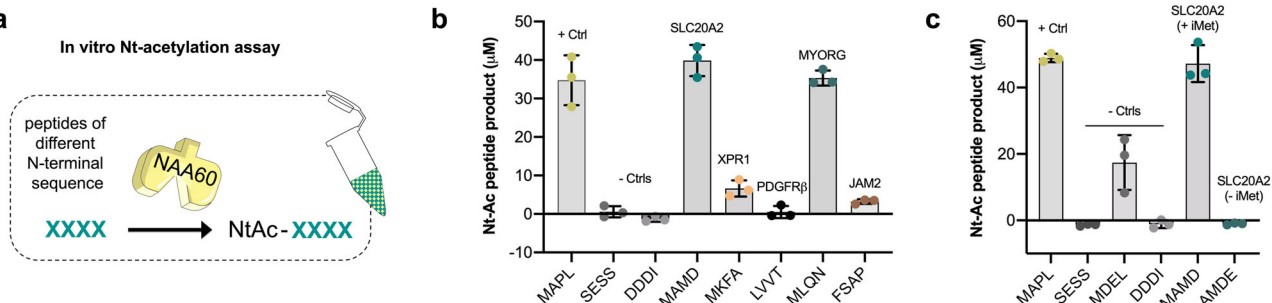

**Fig. 5 | Nt-acetylation assay of peptides involved in PFBC shows that SLC20A2 is the best substrate among them. a** Schematic representation of the in vitro DTNB enzyme activity test towards peptides representing the N-termini of PFBC-connected proteins. NAA60 was expressed and purified from *E. coli* and mixed with various peptides in acetylation buffer. **b** NAA60 was tested towards the classic NAA60 substrate MDEL as a positive control as well as classic NAA10 and NAA80 peptides as negative controls alongside the N-termini of PFBC-connected proteins. All peptides, except MAMD and MLQN, were significantly different from the positive control by two-tailed *t* test (*p* values ranging from 0.0088 to 0.011, exact *p* values are provided in the Source Data file). Mean from *n* = 3 independent

experiments is shown, and error bars represent SD. **c** NAA60 was tested towards the classic NAA10, NAA20, NAA80 and NAA60 substrates in addition to full and potential iMet-processed SLC20A2 N-termini. All peptides, except the MAMD (SLC20A2), were significantly different from the positive control by two-tailed *t* test (*p* values ranging from 0.0001–0.02, exact *p* values are provided in the Source Data file). Mean from *n* = 3 independent experiments is shown, and error bars represent SD. Parts of the figure were drawn by using elements modified with text, markings, and annotations from Servier Medical Art, provided by Servier, licensed under a Creative Commons Attribution 3.0 (https://creativecommons.org/licenses/by/3.0/).

NAA60 or NatF is part of a protein family of seven human N-terminal acetyltransferases (NATs). Each human NAT catalyses Nt-acetylation of distinct substrates[19,34]. While NatA–NatE perform co-translational N-terminal acetylation, NatF and NatH post-translationally Nt-acetylate membrane proteins and actin, respectively. The NATs Nt-acetylate ~80% of the human proteome[16,35] with most events driven by NatA, NatB, and NatC[16,36,37]. The functional impact of Nt-acetylation varies from protein to protein and may involve protein folding, degradation, stabilization, subcellular targeting, and protein complex formation[19,34]. Recent data strongly suggest that a major function of Nt-acetylation is shielding proteins from proteasomal degradation[38–40]. Interestingly, NatC Nt-acetylates a subset of proteins starting with Met (Met-Leu-, Met-Ile, Met-Phe, Met-Trp, Met-Val, Met-Met, Met-Lys etc.)[37,41–43], and for many of these proteins, lack of NatC-mediated Nt-acetylation will cause them to be recognized by specific ubiquitin E3 ligases and degraded[38] in the N-degron pathway[44]. NatC and NatF have distinct in vivo substrates since NatC acts co-translationally on a variety of proteins while NatF acts post-translationally on transmembrane proteins, but these two enzymes acetylate similar types of protein N-terminal sequences[34]. Therefore, it is possible that NatF/NAA60 transmembrane protein substrates with diminished Nt-acetylation are also recognized and degraded by the N-degron pathway. Here, we show that *NAA60*-related disease mechanisms involve impaired cellular phosphate homeostasis, possibly via SLC20A2.

Variants in five NAT genes have been associated with a human phenotype, *NAA10 (OMIM:3000013), NAA15(OMIM:608000),*

*NAA20 (OMIM:610833), NAA30 (OMIM:617989),* and *NAA80 (OMIM: 607073)*. Variants in *NAA10*, encoding the catalytic subunit of NatA, were associated with aged appearance, craniofacial anomalies, hypotonia, global developmental delay, cryptorchidism, and cardiac arrhythmias[45]. Partially overlapping phenotypes were observed in individuals with pathogenic variants in NAT genes, including developmental delay, intellectual disability, congenital heart disease and autism spectrum disorder (*NAA10* and *NAA15*)[46–51], developmental delay, intellectual disability, and microcephaly (*NAA20*)[52,53], developmental delay (*NAA30*)[54], hearing loss, muscle weakness and developmental delay (*NAA80*, the actin-specific NAT)[55,56].

In this work, we present pathogenic *NAA60* variants as causative for PFBC with impaired cellular P$_i$ homeostasis, possibly via SLC20A2 (Fig. 7). Moreover, brain calcifications are frequent in the general population and increase with age from ~1% in young adults to ~20% in the elderly[28,57]. They are also found in a variety of major neurodegenerative conditions. Almost half of Down syndrome cases[58], 20% of Parkinson's disease (PD)[59], and 10% of Alzheimer's disease (AD)[60] cases develop brain calcifications with overlapping histological pathogenesis and clinical manifestations[58,60]. As such, deciphering the genetic basis and molecular mechanisms of brain calcification may facilitate a better understanding of disease processes associated with common neurodegenerative disorders.

## Methods
The individuals included in this study were recruited along with unaffected family members with informed consent under ethics-approved

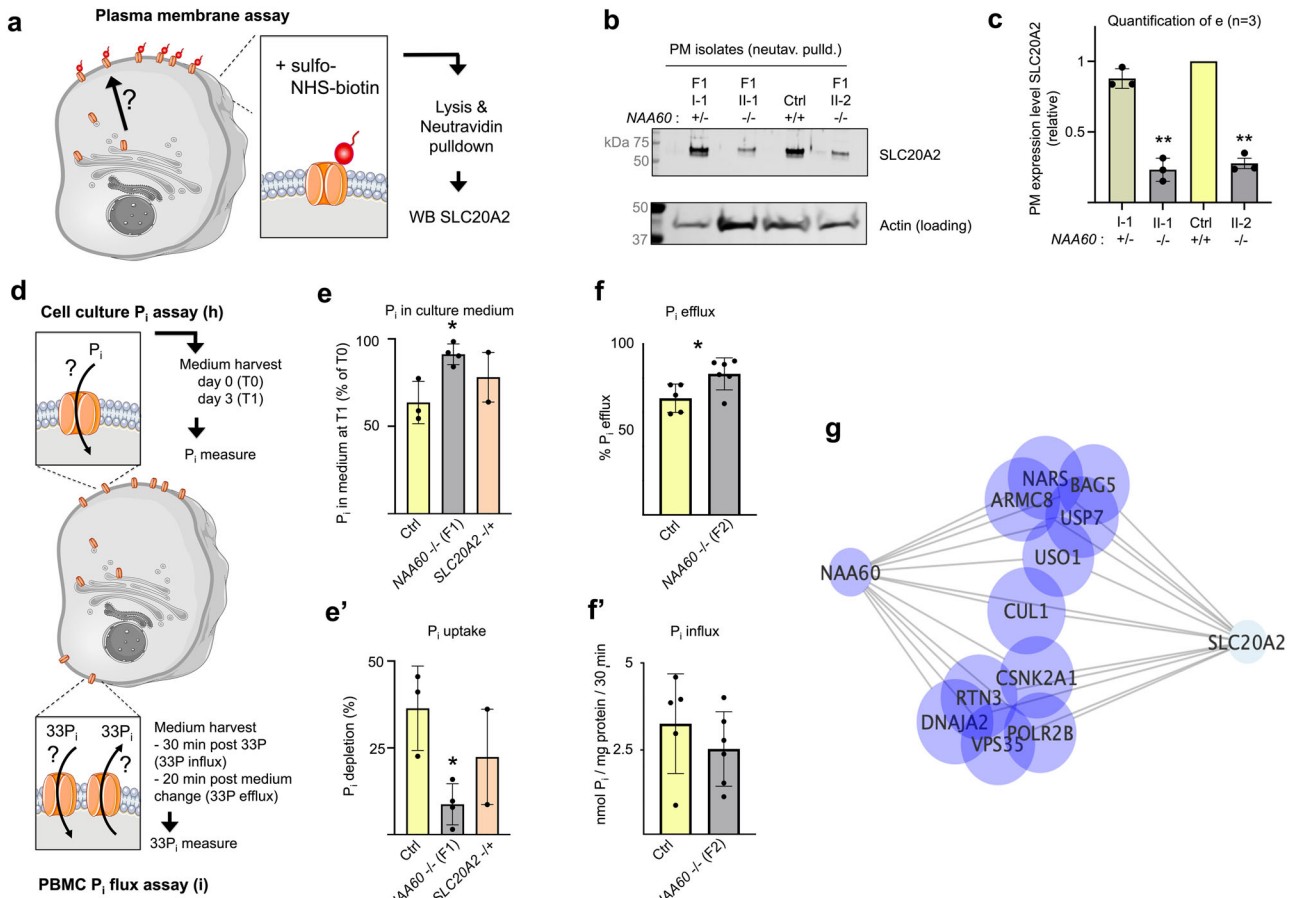

**Fig. 6 | *NAA60*-related disease mechanisms involve impaired cellular phosphate homeostasis, possibly via SLC20A2. a** Representation of sulfo-NHS-biotin labeling and pulldown to isolate plasma membrane (PM) proteins. **b** Western blot from human fibroblasts subjected to sulfo-NHS-biotin labeling and pulldown. **c** Quantification of **b**, in affected cases (F1-II-1 and F1-II-2), heterozygous unaffected carriers (F1-I-1), and control lines (** indicates $p < 0.0005$ by unpaired two-tailed $t$ test with Welch's correction ($p = 0.0038$ for case I against control and $p = 0.0026$ for case II against control). The results were compared to endogenous actin levels. Mean from $n = 3$ independent pulldowns and error bars represent SD.
**d** Representation of the assay assessing depletion of inorganic phosphate ($P_i$) from culture medium. **e** Quantification of $[P_i]$ in culture medium of fibroblasts from cases with variants in *NAA60*, *SLC20A2,* and healthy controls after 3 days of incubation (T1). Mean from $n = 3$ independent experiments shown, error bars represent SD, and * indicates $p < 0.05$ by unpaired two-tailed $t$ test with Welch's correction ($p = 0.0423$). **e'** $[P_i]$ depletion from T0 to T1 in fibroblast culture as in **e**, showing higher depletion in control (43.3%) than in *NAA60* mutant culture medium (7.9%),

indicating a 35.5% reduction in $[P_i]$ uptake by *NAA60* mutant fibroblasts. Mean from $n = 3$ independent experiments is shown, error bars represent SD, and * indicates $p < 0.05$ by unpaired two-tailed $t$ test with Welch's correction ($p = 0.0423$). **f** Efflux of $^{33}P_i$ and **f'** uptake of $^{33}P_i$ from in PBMCs in healthy donor (Ctrl) and PFBC patient from F2 carrying the *NAA60* c.338-1 G > C variant. Mean and SD from $n = 6$ independent experiments are shown, and * indicates $p < 0.05$ by unpaired two-tailed $t$ test with Welch's correction ($p = 0.0253$). **g** *NAA60* and *SCL20A2* genes module co-expression in the caudate tissue. Top–down plot of the blue and gray module genes in the caudate tissue. *NAA60* module is highlighted in blue, *SLC20A2* in gray. Size of gene nodes reflect their connectivity with the rest of genes in the module. Proximity of genes in the plot reflects their similarity in terms of shared connections with other genes. Error bars show SD. * indicates $p < 0.05$ by unpaired $t$ test with Welch's/ correction. Source data are provided in the Source Data file. Parts of the figure were drawn by using elements modified with text, markings, and annotations from Servier Medical Art, provided by Servier, licensed under a Creative Commons Attribution 3.0 (https://creativecommons.org/licenses/by/3.0/).

---

research protocols (UCLH: 04/N034; Rouen: RBM-0259 approved by the CPP Ile de France II ethics committee, and CERDE (notification E2023-40, Rouen University Hospital) approved the retrospective analysis of genomic and phenotypic data; Comité de Protection des Personnes ID-RCB/EUDRACT: 2014-A01017-40); Saudi Arabia: KFSHRC IRB (RAC# 2121053)). The 100,000 Genomes Project Protocol has ethical approval from the HRA Committee East of England−Cambridge South (REC Ref 14/EE/1112). The authors affirm that the human research participants' guardians provided written informed consent for the publication of the video in Supplementary Movie 1. Human research participants/their guardians provided written informed consent for generation of fibroblast cell lines from skin biopsies.

## Study participants, phenotype, and clinical measures
Patients with a negative family history or recurrence among siblings and confirmed clinical and radiological diagnosis of PFBC were

recruited from multiple centers in the UK (UCL series). In this series, we screened 78 cases from 53 families with PFBC who were negative for pathogenic variants in *SLC20A2, PDGFB, PDGFRB, XPR1, MYORG, JAM2,* and *CMPK2*. Secondary causes of brain calcification were excluded in all cases. Neurogenetics specialists performed comprehensive neurological examination and phenotyping.

Following the identification of a family linked to biallelic *NAA60* variants in the UCL series, we screened additional cases from two French centers. At the Rouen University Hospital and Inserm U1245 (Rouen, France), we screened exome sequencing data from 102 unrelated probands with no causative variant in any of the seven known PFBC genes. Patients were recruited through the French PFBC study as previously described[10]. At the laboratory of Neurogenetics and Neuroinflammation, Institut Imagine, Paris, we screened 30 unrelated families with unexplained brain calcifications. Then, we screened the Solve-RD-Connect genomics platform containing

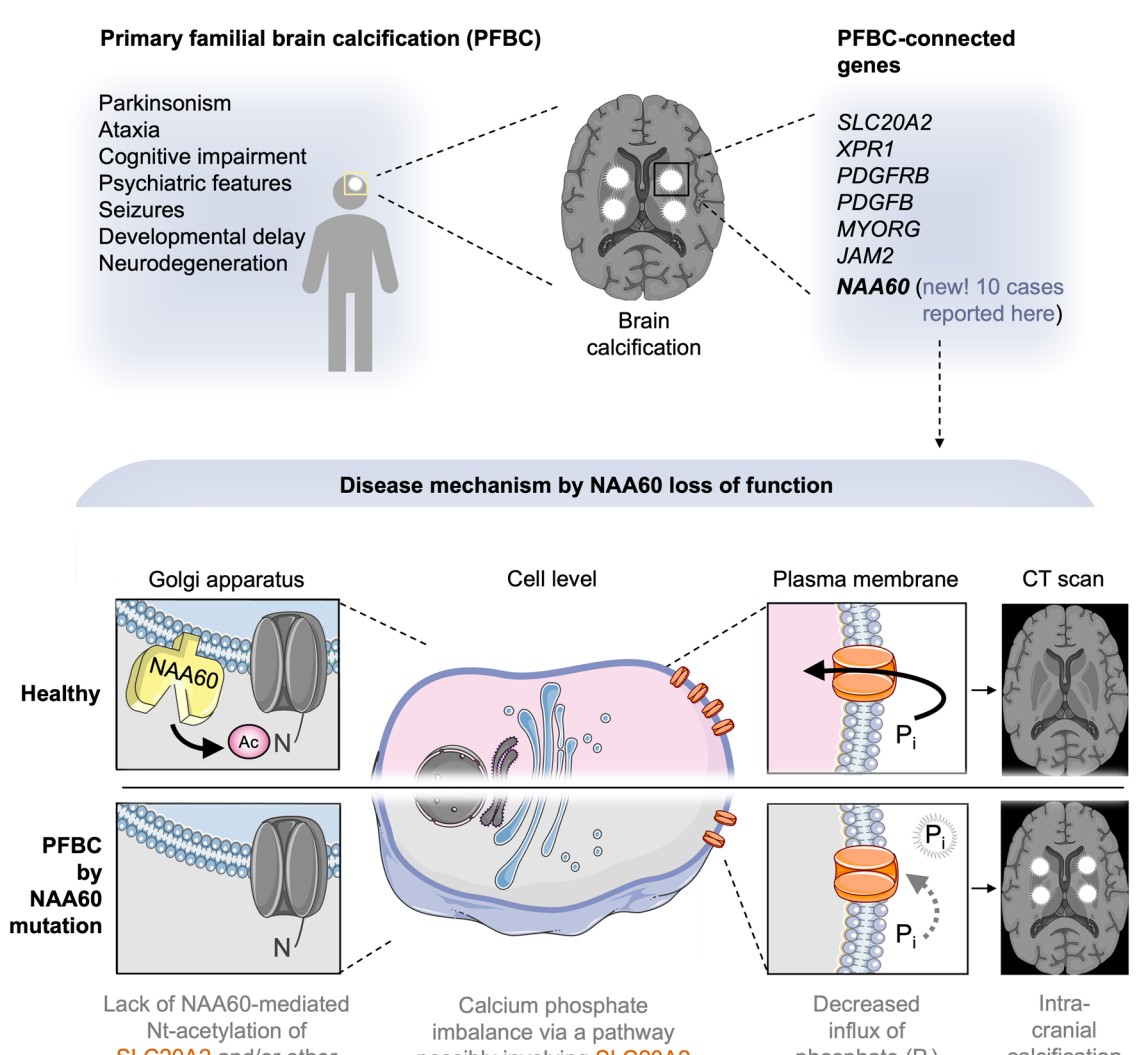

**Fig. 7 | Illustration summarizing the key findings of this work and the possible *NAA60*-related disease mechanism.** *NAA60* is established here as a PFBC-linked gene by loss of its N-terminal acetyltransferase activity in cases homozygous for *NAA60* variants. A possible molecular mechanistic path through which loss of NAA60 function may cause primary familial brain calcifications (PFBCs) is shown. Loss of NAA60 function entails a lack of N-terminal acetylation (Ac-N) of several membrane proteins at the Golgi apparatus, possibly including some with critical function in brain phosphate homeostasis, such as SLC20A2. The impaired targeting or function of these non-Nt-acetylated NAA60 substrates may lead to the build-up of phosphate and calcium phosphate in the brains of patients with loss-of-function variants in *NAA60*, possibly via a pathway involving the phosphate importer SLC20A2. Parts of the figure were drawn by using elements modified with text, markings, and annotations from Servier Medical Art, provided by Servier, licensed under a Creative Commons Attribution 3.0 (https://creativecommons.org/licenses/by/3.0/).

shared genome-phenome datasets from 22,000 individuals with whole-exome sequencing data, including affected participants and relatives, of which 13,676 were affected participants with rare diseases[21]. Then, we screened WGS data from 75,880 participants in the Rare Disease arm of the 100,000 Genomes Project[61]. These comprised 35,451 rare disease probands with 6139 affected family members and 13,920 participants (offspring) for whom trio WGS data were available. Finally, we screened two series of unsolved rare disease cases from *Centogene* and Saudi Arabia. The Saudi Arabia cohort, consisted of ten cases with suspected genetic stroke, which were analyzed using WES that revealed one *NAA60* variant, five cases were linked to known disease genes, and the remaining four were submitted for reanalysis.

In cases with biallelic *NAA60* variants, the results from additional investigations were retrospectively analyzed based on chart review, where available. Neuroimaging with computed tomography (CT) and brain MRI confirmed the presence of brain calcifications in all reported adult cases (*n* = 9). Cognitive impairment was assessed by formal psychometry, when available.

We used the term sex (biological attribute) for reporting the findings. Sex was the only information that was collected and was determined on self-report. We reported here only the number of male versus female. No disaggregated sex analysis was performed as the sample size was too low to enable meaningful conclusion. Sex was not considered in the study design. Age was defined by the data reported by patient or their carer. Age information was used to establish disease onset, disease duration, and time to brain imaging. Genotypic information was used to correlate with clinical and brain neuroimaging features. Neurological examination was described including motor features, cognitive features, psychiatric features.

**Genome-wide sequencing**
DNA was extracted from peripheral blood in all cases. WES was performed in all families except Family 4, which had WGS performed. Illumina HiSeq4000 (in F1, F3, and F7), Illumina HiSeq2500 (in F4), and Illumina HiSeq2000 (in F2, F5, and F6) (Illumina, San Diego, CA, USA) sequencers were used to generate 100 bp or 150-bp paired-end reads. Alignment was performed using BWA[62] with GRCh38 as a reference.

Variants were called using the GATK[63–66] Unified Genotyper-based pipeline[63–65] workflow. All variants were annotated using ANNOVAR[67] and filtered using custom R scripts. Only novel or very rare variants with a minor allele frequency (MAF) of <0.01 in the genome Aggregation Database v2.1 (gnomAD)[68] were included. Variants were filtered for biallelic, highly deleterious, rare variants segregating with the disease in the trios. The WGS and bioinformatic pipelines, as well as the "tiering" framework for variant prioritization for cases included in the 100,000 Genomes Project, have been previously described[61]. Structural variants detected in 33,924 families with WGS data were prioritized using a MySQL database of variants called by Manta and CANVAS, as described[69]. For the *NAA60* (ENST00000407558, GenBank transcript ID NM_001083601), variant segregation was confirmed in all other family members, and primer sequences are as follows: c.321_327del, forward: gacgtgtgagcctgactttc, reverse: ggaagagactgggtgcagat; c.338-1 G > C, forward:GTGGTGGTGAAGAAAGGCTG, reverse: GTGTTGTTGGTGGTGAGGAC; c.391 C > T, forward: AAGTTTAAAGGATCACATATCAACCA, reverse: TGCTGCTTGAAGTCTCTGTTTT; c.130 C > T, forward: gtggcaggttcacctgagta, reverse: TTGGTCCTGTTCTTAATTTCAGC; c.50 T > G, forward: GTGTGAATGACAGAGGTGGTG, reverse: ACCAGTCGCCACACAGGT; c.428 A > C, forward: TCATCCTTCCTTCCTAACTGCTC, reverse: TTAAGGATATAAAATCGTCCAGGG.

## Haplotype analysis

For relatedness analysis, samples from F1, F2, and F3 were realigned to GRCh38, and variants were called using the GATK HaplotypeCaller-based pipeline[50,64,65]. Shared regions of homozygosity were identified using HomozygosityMapper[70]. Kinship and relatedness were further inspected with Peddy[23] and Automap[24].

## RNA analysis

To evaluate the c.338-1 G > C variant in intron 5, a region flanking exon 6 was amplified by PCR from the proband from F2 and a control individual with specific primers containing an additional *Xho*I and *Bam*HI restriction site (forward primer with *Xho*I restriction site: 5′-aattctcgagATTTTAAACCCAGGCCCATC-3′ and reverse primer with *Bam*HI restriction site: 5′-attggatccGTGCAACTCGATTTACCCAAA-3′) with a modified protocol[71,72]. After PCR amplification and clean-up, restriction enzyme digestion of the PCR fragment and pSPL3 exon trapping vector was performed prior to ligation between exon A and exon B of the linearized pSPL3 vector. The variant-containing and wild-type vectors were transformed into DH5α competent cells (NEB 5-α, New England Biolabs), plated, and incubated overnight. The wild-type and mutant-containing vector sequences were confirmed by Sanger sequencing.

Vectors containing homozygous and wild-type sequences were transfected into HEK293T cells (ATCC) at a density of $2 \times 10^5$ cells per mL. One microgram of the respective pSPL3 vectors was transiently transfected using 6 μl of FuGENE 6 Transfection Reagent (Promega). An empty vector and transfection negative reactions were included as controls. The transfected cells were harvested after 24 hours. Total RNA was prepared using a miRNAeasy Mini Kit (Qiagen). Approximately 1 μg of RNA was reverse transcribed using a High-Capacity RNA-to-cDNA Kit (Applied Biosystems) according to the manufacturer's protocols. The cDNA was PCR amplified using vector-specific SD6 forward (5′-TCTGAGTCACCTGGACAACC-3′) and exon 6-specific (5′-CATGCAGGTAAATGGCTTTGC-3′) primers. The amplified fragments were visualized on a 1% agarose gel and Sanger sequenced.

## Generation of patient-derived cells

Sub-deltoid skin punch biopsies were used to generate primary fibroblast cell lines from cases (F1-II-I, F1-II-2), unaffected relatives(F1-I-1), and healthy controls. The area for biopsy was scrubbed with 70% isopropyl alcohol, and 2% lidocaine local anesthetic was administered prior to removal of a 3 mm diameter skin sample by a sterile punch biopsy instrument. The skin wound was closed with steristrips. Skin punch biopsies were used to generate primary fibroblast cell lines by manual and enzymatic dissociation, followed by growth in culture media.

## Construction of plasmids for the expression of NAA60 WT and NAA60 variants

For the generation of the FLAG-tagged *NAA60* WT construct, *NAA60* cDNA clones were subcloned into SNAP-DHHC3 plasmids [removing SNAP-DHHC3] (NEB, Ipswich, MA, USA) and conjugated to a FLAG tag (Sigma Aldrich, St. Louis, MO, USA). The *NAA60* F1 construct, in which a premature stop codon was inserted at amino acid 108, was generated by PCR of *WT* cDNA. Constructs representing *NAA60* variants from F2/3, F4, F5, and F6 were generated from the WT or F1 constructs by mutagenesis (Q5® Site-Directed Mutagenesis Kit, NEB). Constructs representing *NAA60* variants from F7 were generated from the WT construct by mutagenesis (GeneArt™ Site-Directed Mutagenesis System, Thermo Fisher Scientific). For purification of NAA60 from *E. coli*, SUMO-NAA60$_{1–184}$ was used[26].

## Cell culture and transfection

HeLa cells for NAA60 WT localization experiments were maintained at 37 °C in 5% $CO_2$ in Dulbecco's modified Eagle's medium (DMEM) (Gibco, Grand Island, NY, United States) supplemented with 10% fetal bovine serum (FBS) (Gibco). Plasmid transfections were performed with Fugene HD (Promega, Madison, WI, USA) as recommended by the manufacturer. Transfection of cDNA was performed at a final concentration of 100 μg/ml. Human fibroblasts from patients, carriers, and healthy controls were maintained at 37 °C in 5% $CO_2$ in fibroblast medium DMEM (Sigma, 6546) supplemented with 10% FBS, 1% L-glutamine, 1% antibiotic-antimycotic solution, 0.1% fibroblast growth factor, and 0.1% hydrocortisone (Cell Biologics, Chicago, IL, USA).

RPE-1 cells (CRL-4000, ATCC) used for NAA60 subcellular localization studies of FLAG-NAA60 WT and variants and HEK293T cells (12022001, Sigma) used for immunoprecipitation of the same proteins were cultured in DMEM and high glucose with pyruvate (Sigma, 6546) supplemented with 10% FBS and 1% penicillin/streptomycin and maintained as described above. HAP1 cells were obtained from Horizon Genomics and cultured as recommended. Parental control (C631), *NAA60* knockout (HZGHC003172c010), and NAA80 knockout (HZGHC003171c012) HAP1 cells were grown in Iscove's modified Dulbecco's medium with 10% FBS and 1% penicillin/streptomycin. Since HAP1 cells possess an originally near-haploid genome but spontaneously switch to a diploid state during normal cultivation, all HAP1 cell lines were passaged until diploidy could be confirmed using flow cytometry recommended in the HAP1 cell culture guidelines by Horizon as well as detailed in a methods article on the procedure[27].

Plasmid transfection of RPE-1 and HEK293T cells was performed with XtremeGene 9 (Roche) according to the recommended protocol and using 2 μl transfection reagent per μg plasmid. For microscopy, transfected cells were fixed approximately 18 hours post transfection; for immunoprecipitation, cells were harvested 48 hours post transfection.

## Immunofluorescence and fluorescence microscopy

For NAA60 WT imaging experiments, transfected HeLa cells were chemically fixed in 4% paraformaldehyde in phosphate-buffered saline (PBS) for 10 min at room temperature (RT). Cells were then permeabilized in permeabilization buffer (0.3% NP-40, 0.05% Triton-X-100, 0.2% bovine serum albumin (BSA) (IgG free), 1× PBS) for 3 min at RT, washed three times in wash buffer (0.05% NP-40, 0.05% Triton-X-100, 0.2% BSA (IgG free), 1× PBS), and subsequently blocked in blocking buffer (0.05% NP-40, 0.05% Triton-X-100, 5% normal goat serum, 1× PBS) for 1 hour. Primary antibodies were diluted at 1:500 in blocking

buffer and added to cells for 1 hr at RT. Next, the cells were washed three times in wash buffer (0.05% NP-40, 0.05% Triton-X-100, 0.2% BSA (IgG free), 1× PBS), incubated with a dilution of a fluorophore-labeled secondary antibody (1:1000) in blocking buffer for 1 hour at RT, washed three times with wash buffer, and twice in PBS. To investigate the distribution of a wild-type NAA60 construct conjugated to a FLAG epitope tag in HeLa cells, we used structured illumination microscopy (SIM). For SIM samples, the cells were mounted with ProLong Gold antifade reagent (Life Technologies, Carlsbad, CA) and 1 mm thick precleaned microscope slides (Thermo Fisher Scientific, Waltham, MA). A final super-resolved image was reconstructed by super-imposing several raw images at various diffraction grating orientations.

For imaging comparing the localization of FLAG-NAA60 WT to mutated F1-F5 variants, confocal images were obtained using a Leica TCS SP8 STED 3× confocal laser microscope equipped with an HC PL APO 100 × 1.4 NA oil objective, 1 Airy unit pinhole aperture. Images were acquired as Z-stacks with the Leica Application Suite X software applying the Lightening module, exported, and processed in ImageJ to show Z-projections of merged stacked planes.

The antibodies used for imaging were rabbit anti-GPP130 (Covance, PRB144C), rabbit anti-GM130 [EP892Y] (Abcam, ab52649), and mouse anti-FLAG M2 (Sigma, F1804). All secondary antibodies conjugated to fluorescent dyes were purchased from Molecular Probes (Eugene, OR, USA).

### Surface biotinylation and western blot

Patient fibroblasts were grown in T75 flasks to 100% confluency. The fibroblasts were washed twice in PBS and subsequently incubated in 5 ml/flask of a 0.5 mg/ml sulfo-NHS-biotin solution in PBS (Thermo-Fisher) for 30 min on ice. The cells were then washed in quenching buffer (PBS, 100 mM glycine) and incubated in a 10 ml/flask for 15 min on ice. The cells were washed once with PBS, scraped, incubated in lysis buffer (50 mM HEPES-NaOH, pH 7.4, 100 mM NaCl, 5 mM EDTA, (v/v) Triton-X-100, protease inhibitor cocktail), sonicated, and subjected to pulldown with Neutravidin-agarose (Pierce). Inputs were compared to endogenous actin. Levels of biotinylated surface proteins were estimated based on a ratio of control to patient fibroblasts. The protein concentration in the lysate was monitored by a Bradford assay, in which the total protein concentration was determined at 595 nm according to the manufacturer's protocol for a standard 1-ml cuvette assay (Bio-Rad, 500-0203, Mississauga, ON). Samples were analyzed by standard Western blot. The antibodies used were rabbit anti-SLC20A2 (Abcam, ab191182) and rabbit anti-actin (Sigma, A2066). As the immunogen used to generate the SLC20A2 antibody maps to a region of SLC20A2 that has some sequence match with SLC20A1, it cannot be excluded that this antibody could also have reactivity towards SLC20A1, which has a very similar molecular weight. Antibodies used for other Western blots were mouse anti-FLAG M2 (Sigma, F1804) used at 1:1000, mouse anti-GAPDH (Santa Cruz Biotechnology, Sc-47724) used at 1:5000, rabbit anti-NAA60 polyclonal (BioGenes custom made Arnesen lab) used at 1:500. Secondary antibodies for western blot were ECL™ Anti-Rabbit IgG HRP Linked Whole antibody (Cytiva Life science, NA934) used at 1:3000 and ECL™ Anti-Mouse IgG HRP Linked Whole antibody (Cytiva Lifescience, NA931V) used at 1:3000.

### DTNB in vitro peptide Nt-acetylation assay for NAA60 purified from E. coli

Purification of His-SUMO-hNAA60$_{1-184}$ was performed as previously described[73] and detailed here. The plasmid was transformed into *E. coli* BL21 (Invitrogen) by heat shock. The bacteria were grown in 200 ml of LB at 37 °C until an OD$_{600}$ of 0.6. The culture was kept on ice for 15 min before His-SUMO-NAA60$_{1-184}$ expression was induced by 200 μM IPTG. After 20 hours of incubation at 17 °C, the bacterial culture was harvested, and the pellet was stored at −20 °C. The bacterial pellet was

thawed on ice, and the cells were lysed by sonication in lysis buffer (100 mM HEPES pH 7.5, 300 mM NaCl, 2 mM DTT supplemented with protease inhibitor cocktail EDTA-free (Roche)). Cell debris was removed by centrifugation at 15,000 × g for 30 min. The supernatant was applied to a 5 ml HisTrap column, the column was washed with wash buffer (100 mM HEPES pH 7.5, 200 mM NaCl, 2 mM DTT, 20 mM imidazole), and the protein was eluted by slowly increasing the imidazole concentration to 300 mM. Eluted fractions containing His-SUMO-NAA60$_{1-184}$ were combined and loaded on a Size Exclusion Chromatography column (Superdex 200 16/600, GE Healthcare). The fraction containing monomeric His-SUMO-hNAA60$_{1-184}$ was collected. Protein purity was confirmed by SDS/PAGE and Coomassie staining, and concentration was determined by absorbance measurements at 280 nm. The purified protein was concentrated by using Amicon Ultra Centrifugation filters and frozen in small aliquots at -80 °C.

For the peptide in vitro Nt-acetylation assay, purified SUMO-NAA60$_{1-184}$ (500 nM) was mixed with substrate peptides (300 μM) and Ac-CoA (300 μM) in acetylation buffer (50 mM HEPES pH 7.5, 100 mM NaCl, 1 mM EDTA) to a total reaction volume of 50 μl and incubated at 37 °C for 20 s. Negative controls without enzyme were also prepared. Acetylation was stopped by adding 100 μl quenching buffer (3.2 M guanidin-HCl, 100 mM Na$_2$HPO$_4$ pH 6.8). For negative controls, enzyme (500 nM) was added after quenching. The samples were mixed with 20 μl DTNB buffer (100 mM Na$_2$HPO$_4$ pH 6.8, 10 mM EDTA) prepared with fresh DTNB (10 mg/ml). Then, 150 μl of each sample was transferred to a 96-well plate, and the absorbance was measured at 412 nm[74]. For each peptide tested, three replicates of positive and negative controls were used. The absorbance of negative controls was averaged and subtracted from the absorbance of the samples. The product concentration was calculated using the Lambert Beer equation and the extinction coefficient of TNB2-(13700/M cm). The following synthetic peptides (>90% purity) were used (all 24-mer oligopeptides from Biogenes): Protein phosphatase 6, O00743, MAPL, verified NAA60/NatF substrate[18]: [H] MAPLDLDRWGRPVGRRRRPVRVYP [OH]; High mobility group protein A1, P17096, SESS, verified NatA substrate[35]: [H] SESSSKSRWGRPVGRRRRPVRVYP [OH]; NF-кkB p65, Q04206, MDEL, verified NatB substrate[75]: [H] MDELFPLRWGRPVGRRRRPVRVYP [OH]; mature β-actin, P60709, DDDI, verified NAA80/NatH substrate[56]: [H] DDDIAALRWGRPVGRRRRPVRVYP [OH]; SLC20A2, Q08357, MAMD or the potential iMet-processed AMDE: [H] MAMDEYLRWGRPVG RRRRPVRVYP [OH] and [H] AMDEYLRWGRPVGRRRRPVRVYP [OH] and XPR1, Q9UBH6, MKFA:[H]-MKFAEHLRWGRPVGRRRRPVRVYP-[OH]; and from Innovagen: PDGFRB, P09619, MRLP: [H]-LVVTPPG RWGRPVGRRRRPVRVYP-[OH], MYORG, Q6NSJ0, MLQN: H]-MLQNPQ ERWGRPVGRRRRPVRVYP-[OH], JAM2, P57087, FSAP: H-FSAPKDQRW GRPVGRRRRPVRVYP-[OH].

### Immunoprecipitation and [14-C]-Nt-acetylation assay

To obtain NAA60 protein variants for in vitro testing, these were expressed and immunoprecipitated from HEK293T cells. A total of 4 × 10$^6$ HEK293T cells/dish were seeded on 10-cm dishes and grown for 24 h before transfection. Forty-eight hours after transfection, the cells were washed in cold PBS, harvested in cold PBS by centrifugation at 17,000 × g for 1 min, and frozen at −80 °C. Pellets were thawed, resuspended in 2× NAA60 acetylation buffer (100 mM Hepes pH 7.5, 200 mM NaCl, 2 mM EDTA, Complete EDTA-free protease inhibitor (Roche)), and 20 μl buffer/mg pellet), and tubes were left on a rotating wheel for 25 min. The lysates were given a short spin (15 sec) on a bench top centrifuge, and 2 μg anti-FLAG (Sigma F1804-20046) was added to the supernatant. The mixture was incubated on a rotating wheel for 3 h before 200 μl prewashed magnetic beads (Invitrogen Dynabeads 10004D) were added and again incubated on a rotating wheel overnight. The beads were washed three times in IPH lysis buffer and one time in 2× acetylation buffer (100 mM HEPES pH 7.5, 200 mM NaCl, 2 mM EDTA). Stability tests showed that transfections with FLAG-

NAA60-F1 and FLAG-NAA60-F2/3 produced drastically lower output (Fig. 2c). Therefore, to obtain enough protein for these two variants to be tested in an in vitro activity assay next to the WT, several adjustments were made, including increased plasmid amounts, an increased number of transfected cells, and MG132 treatment for 6 hours prior to harvest (only for F2/3). All mutants were tested strictly in parallel with the WT, and a final adjustment was performed by normalizing enzyme inputs according to parallel Western blots.

For the [14-C]-Nt-acetylation assay, magnetic beads from FLAG-NAA60 IP, [14-C]Ac-CoA (25 μM), and 200 μM substrate peptide MAPL ([H] MAPLDLDRWGRPVGRRRRPVRVYP [OH]) were mixed in acetylation buffer (50 mM HEPES pH 7.5, 100 mM NaCl, 1 mM EDTA) and incubated at 37 °C on a shaker for 2 h. Beads were isolated using a magnet, and 20 μl of the suspension was transferred to P81 phosphocellulose paper. After air-drying, the filters were washed 3× for 5 min in 10 mM HEPES pH 7.4. The filters were again air-dried and transferred to scintillation tubes, and 5 ml scintillation solution was added before counting in a scintillation counter. For each condition, three replicates were performed, with pretests performed to be able to apply equal enzyme inputs.

## Quantification of inorganic phosphate

We quantified the extracellular free inorganic phosphate ($P_i$) in fibroblast cultures from F1 by using the $P_i$Per Phosphate Assay kit (Life Technologies, Grand Island, NY) according to the manufacturer's instructions. Fibroblast cell lines from affected patients, carriers, and healthy controls were cultured under standard conditions. The culture medium was collected after three days of incubation as previously described[76]. All assays were performed in triplicate in clear bottom 96-well assay plates (BD Falcon). All plates were incubated at 37 °C in the dark for 15 s and read in an iMarkTM Microplate Absorbance Reader (Bio-Rad Laboratories) at excitation/emission wavelengths of 544/590 nm.

Peripheral blood mononuclear cells (PBMCs) of one patient carrying the NAA60 c.338-1 G > C variant from F2 and one sex- and age-matched control were obtained from fresh blood samples. Phosphate uptake and efflux assays in PBMCs were performed similar as in earlier studies[31,77]. For uptake measurements, cells were incubated for 30 min at 37 °C in phosphate-free DMEM supplemented with 0.5 μCi/ml $^{33}$P (FF-1; Hartmann Analytic GmbH, Germany). Cell lysates were then assayed for radioactivity by scintillation counting and for protein content by the BCA protein assay (Pierce). Phosphate uptake was calculated from the concentration of cold phosphate in the medium multiplied by the ratio of cellular [$^{33}$P]phosphate to total [$^{33}$P]phosphate supplemented within a period of 30 s. For the efflux measurements, cells were incubated for 20 min at 37 °C in phosphate-free DMEM supplemented with 0.5 μCi/ml $^{33}$P, gently washed three times with phosphate-free medium, and then incubated in DMEM for 30 min at 37 °C in 10 mM phosphate before collection of the supernatant. The amount of $^{33}$P in the supernatants and cell lysates was measured by liquid scintillation. Phosphate uptake was calculated from the concentration of cold phosphate in the medium multiplied by the ratio of cellular [$^{33}$P]phosphate to total [$^{33}$P]phosphate supplemented (FF-1; Hartmann Analytic GmbH, Bern Switzerland) within a period of 30 s. The percentage of phosphate efflux was calculated as the ratio of released [$^{33}$P]phosphate to total cellular [$^{33}$P]phosphate.

## Co-expression networks analyses

To identify modules in co-expression networks including NAA60 and SLC20A2, we created a co-expression network by using Weighted Gene Co-expression Network Analyses (WGCNA) on GTEx V6 bundle. Gene modules within co-expression networks represent groups of genes that work together on the corresponding samples and are often enriched in specific pathways or with molecular functions. First, we constructed co-expression networks for NAA60 and SLC20A2 using genes with RPKM values above 0.1 in at least 80% of the samples, within the

five relevant brain areas involved in the human phenotype with available data from GTEX (cerebellum, substantia nigra, putamen, nucleus accumbens and the caudate)[78]. The expression levels of each module were summarized by the first principal component (the so-called module eigengene). The network is programmatically accessible from CoExpGTEx (https://github.com/juanbot/CoExpGTEx) and through the Web at CoExpWeb software tool (https://rytenlab.com/coexp). Briefly, we selected genes based on their expression abundance, using only genes with RPKM values above 0.1 in more than 80% of samples. We corrected for batch effect using ComBat[79] and regressed for RNA integrity numbers (RIN), post-mortal interval (PMI), age and gender and used the corresponding residuals to construct the network using CoExpNets R package[80], based on the WGCNA R package[81]. Briefly, we obtained the human phenotype (HP) terms enrichment for blue and grey60 modules (see Results) using gProfileR[82]. The cell maker enrichment is the original from CoExpGTEx based on a Fisher's Exact test with the genes in the module and manually curated cell marker gene lists from WGCNA and our group. The module membership (MM) of each gene in its module is calculated using the Pearson correlation of the gene expression values with the 1st eigen vector of the whole module gene expression.

The network diagrams were generated with Cytoscape 3.10 by creating the bottom-up plot of the network context of NAA60 and SCL20A2 taken together for the co-expression network of caudate, based on adjacency values. The algorithm put the two genes in an empty canvas using them as seed genes. Iteratively, it adds to the canvas the most connected gene to NAA60 across the whole network, and the most connected gene to SLC20A2. It then adds the corresponding edges between the two pairs of genes. Therefore, both NAA60 and SCL20A2 receive connections from approximately half of the genes in the diagram, respectively. This operation is repeated n times for each seed gene (in this case $n = 7$). If a gene selected for addition to the canvas was already added, then a new gene is selected for addition instead. Note that connection between any pair of genes $g_i$ and $g_j$ is measured as the adjacency between the two genes. Adjacency is a concept used in WGCNA to measure the connectivity of each gene. If two genes show high adjacency, based on Pearson correlation, they are said to be strongly connected. Once the canvas is completed with all genes and links required, the layout of genes is calculated by the edge-weighted spring layout in Cytoscape, based on the Kamada–Kawai algorithm. As a result, genes appearing near to each other, on average, reflect a high adjacency between them. The color of the nodes corresponds to the module name, either blue or gray. Their size reflects the accumulated adjacency for each node (the total sum of adjacencies with the rest of genes in the network). The same approach was applied for NAA60 alone.

## Reporting summary

Further information on research design is available in the Nature Portfolio Reporting Summary linked to this article.

## Data availability

All data generated or analyzed during this study and used to produce the plots are included in this published article (and its supplementary information files). The raw DNA sequencing data from the cases reported here, CT and brain MRI, are protected and are not available due to data privacy laws. Next-generation sequencing (NGS) data can be requested by contacting the corresponding author. Due to ethics restrictions on storing and sharing our patient NGS data, this data will have controlled access and will be limited to individuals who enter a research agreement. Use of this genomic data will be restricted to those named on the agreement, and NGS data will be patient de-identified. Data may be obtained from a third party and are not publicly available. 100,000 Genomes Project data are stored in the National Genomic Research Library (https://doi.org/10.6084/m9.

figshare.4530893.v6), and the RD-Connect Data is stored in the European Genome-Phenome Archive (EGA) and the RD-Connect Genome-Phenome Analysis Platform (GPAP) (https://doi.org/10.1038/s41431-021-00859-0). The following databases were used; human reference genomes hg19 and hg38, Gencode human annotation release 40 (GRCh37), dbSNP 132, 1000 Genomes Project, NHLBI Go Exome Sequencing Project, gnomAD, The immunofluorescence and fluorescence microscopy, surface biotinylation, DTNB in vitro peptide Nt-acetylation assay for NAA60, immunoprecipitation and [14-C]-Nt-acetylation assay, quantification of inorganic phosphate and co-expression networks analyses data generated in this study are provided in the Supplementary Information and Source Data file. For clinical, genetic, and neuroimaging enquiries, please contact v.chelban@ucl.ac.uk, and for inquiries regarding the protein functional work, please contact henriette.aksnes@uib.no or thomas.arnesen@uib.no. The expected timeframe for responses 5 days. Source data are provided with this paper.

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

## Acknowledgements

The authors would like to thank the participants and their families for their essential help with this work. This research was made possible through access to the data and findings generated by the 100,000 Genomes Project. The 100,000 Genomes Project is managed by Genomics England Limited (a wholly owned company of the Department of Health and Social Care). The 100,000 Genomes Project is funded by the National Institute for Health and Care Research and NHS England. The Wellcome Trust, Cancer Research UK, and the Medical Research Council have also funded the research infrastructure. The 100,000 Genomes Project uses data provided by patients and collected by the National Health Service as part of their care and support. The Solve-RD project has received funding from the European Union's Horizon 2020 research and innovation program under Grant Agreement No 779257. H.H. was supported by the MRC (MR/S01165X/1, MR/S005021/1, MRC ICGNMD), Wellcome Trust (221951/Z/20/Z), GP2 ASAP, Michael J Fox Foundation (MJFF), The National Institute for Health Research University College London Hospitals Biomedical Research Center, Rosetree Trust, MSA Trust, MSA Coalition, Brain Research UK, Sparks GOSH Charity, Muscular Dystrophy UK (MDUK). V.C. was supported by the Association of British Neurologists' Academic Clinical Training Research Fellowship (grant ABN 540868) and The Guarantors of Brain (grant 565908). Members of the Arnesen lab are acknowledged for helpful inputs to this study. T.A. acknowledges support from the Research Council of Norway (Project 249843), the Norwegian Health Authorities of Western Norway (Project F-12540), the Norwegian Cancer Society (Project 171752—PR-2009-0222), and the European Research Council (ERC) under the European Union Horizon 2020 Research and Innovation Program under Grant 772039. Microscopy using a Leica TCS SP8 STED 3x confocal microscope was performed at the Molecular Imaging Center (MIC), Department of Biomedicine, University of Bergen. R.H. is a Wellcome Trust Investigator (109915/Z/15/Z), who receives support from the Medical Research Council (UK) (MR/N025431/1 and MR/V009346/1), the European Research Council (309548), the Newton Fund (UK/Turkey, MR/N027302/1), the Addenbrookes Charitable Trust (G100142), the Evelyn Trust, the Stoneygate Trust, the Lily Foundation and an MRC strategic award to establish an International Center for Genomic Medicine in Neuromuscular Diseases (ICGNMD) MR/S005021/1, the NIHR Cambridge Biomedical Research Center (BRC-1215-20014). G.N. acknowledges support from the French National Research Agency (CALCIPHOS, ANR-17-CE14–0008–02), Conseil Régional de Haute-Normandie (APERC 2014 no. 2014–19 in the context of Appel d'Offres Jeunes Chercheurs, CHU de Rouen), Région Normandie and the European Union (Recherche Innovation Normandie - RIN 2018; Europe is involved in Normandie with the European Regional Development Fund (ERDF)). Part of this work is a collaboration between CEA-DRF-Jacob-CNRGH-CHU de Rouen. J.F.D. acknowledges support for France Génomique National infrastructure, funded as part of the "Investissements d'Avenir" program managed by the AgenceNationale pour la Recherche (contract ANR-10-INBS-09). J.L.B was supported by the French National Research Agency (CALCIPHOS, ANR-17-CE14–0008–01). Y.J.C. acknowledges a state subsidy managed by the National Research Agency (France) under the "Investments for the Future" (ANR-10-IAHU-01) and funding from the MRC (Grant code: MC_UU_00035/11). B.V. acknowledges support from the Deutsche Forschungsgemeinschaft (DFG) through the Multiscale Bioimaging Cluster of Excellence (MBExC) and DFG VO 2138/7-1 grant 469177153. BV is a member of the European Reference Network on Rare Congenital Malformations and Rare Intellectual Disability (ERN-ITHACA) (EU Framework Partnership Agreement ID: 3HP-HP-FPA ERN-01-2016/739516). NIH R35 GM142433 to A.M.E. The views expressed are those of the authors and not necessarily those of the NIHR or the Department of Health and Social Care.

## Author contributions

V.C. conceived the study. V.C., H.A., T.A. and H.H. jointly designed the study. V.C., H.A., L.C.L., L.S., P.D., R.M., J.V., D.M., N.M.Z., A-C.R., O.Q., A.B., R.K., V.S., S.E., A.Sc., L.V.S., H.M., A.T., A.T.P., A.P., A.Si., L.S.K., K.M.B., A.K.B., A.C., N.G., AM. DL., R.I., P.B., G.Z., F.S.A., F.Al., H.E.S., F.Ab., JL. B., S.T., A.B., R.O., JF.D., M.A., B.C., B.V., S.G., S.H.K., H.L., S.I. D, K.A.F., N.W.W., R.H., A.M.E., M.M., Y.J.C., G.N. L.G.P., J.B. performed the experiments. V.C., H.A., L.C.L., L.S., P.D., R. M, J.V., D.M. N.M.Z., A-C.R., O.Q., A.B., R.K., V.S., S.E., L.V.S., H.M., A.T., A.T.P., A.P., L.S.K., K.M.B., A.K.B., A.C., N.G., AM. DL., A.Si., R.I., P.B., G.Z., F.S.A., F.Al., H.E.S., F.Ab., JL.B., S.T., A.B., R.O., JF.D., M.A., B.C., B.V., S.G., S.H.K., H.L., S.I D, K.A.F., N.W.W., R.H., A.M.E., M.M., Y.J.C., G.N. L.G.P., J.B. collected and analyzed data. V.C., H.A., prepared the original manuscript draft. V.C., H.A., L.C.L., L.S., P.D., R. M, J.V., D.M. A-C.R., O.Q., A.B., N.M.Z., R.K., V.S., S.E., L.V.S., A.Si., H.M., A.Sc., A.T., A.T.P., A.P., L.S.K., K.M.B., N.G., JL. B., S.T., A.B., R.O., JF.D., M.A., H.E.S., B.C., B.V., S.G., S.H.K., H.L., S.I. D, K.A.F., N.W.W., R.H., A.M.E., J.E.R., M.M., Y.J.C., F.S.A, G.N., T.A., H.H. reviewed and provided editing advised on the final manuscript. All authors discussed the results and implications and commented on the manuscript at all stages.

## Competing interests

P.D. is an orator and scientific board member for Gilead and an orator for Novartis. All other authors declare no competing interests.

## Additional information

Viorica Chelban [1,2,46] ✉, Henriette Aksnes [3,46] ✉, Reza Maroofian [1], Lauren C. LaMonica [4], Luis Seabra [5], Anette Siggervåg[3], Perrine Devic[6], Hanan E. Shamseldin[7], Jana Vandrovcova[1], David Murphy[8], Anne-Claire Richard[9], Olivier Quenez [9], Antoine Bonnevalle[9], M. Natalia Zanetti[10], Rauan Kaiyrzhanov[1,11], Vincenzo Salpietro [1], Stephanie Efthymiou [1], Lucia V. Schottlaender[1,12,13], Heba Morsy [1,14], Annarita Scardamaglia[1], Ambreen Tariq[1], Alistair T. Pagnamenta [15], Ajia Pennavaria[3], Liv S. Krogstad [3], Åse K. Bekkelund[3], Alessia Caiella[3], Nina Glomnes[3,16], Kirsten M. Brønstad[3], Sandrine Tury[17], Andrés Moreno De Luca [18,19], Anne Boland-Auge [20], Robert Olaso[20], Jean-François Deleuze [20], Mathieu Anheim[21,22,23], Benjamin Cretin[21,22,23], Barbara Vona [24,25], Fahad Alajlan [26], Firdous Abdulwahab[7], Jean-Luc Battini [17], Rojan İpek[27], Peter Bauer[28], Giovanni Zifarelli[28], Serdal Gungor[29], Semra Hiz Kurul[30], Hanns Lochmuller [31,32,33], Sahar I. Da'as [34,35], Khalid A. Fakhro [34,35,36], Alicia Gómez-Pascual [37], Juan A. Botía [37], Nicholas W. Wood [8,38], Rita Horvath [39], Andreas M. Ernst [4,40], James E. Rothman [4,10], Meriel McEntagart[41], Yanick J. Crow [5,42], Fowzan S. Alkuraya [7,43], Gaël Nicolas[9], SYNaPS Study Group*, Thomas Arnesen [3,44,47] ✉ & Henry Houlden [1,38,47] ✉

[1]Department of Neuromuscular Diseases, UCL Queen Square Institute of Neurology, London WC1N 3BG, UK. [2]Neurobiology and Medical Genetics Laboratory, "Nicolae Testemitanu" State University of Medicine and Pharmacy, 165, Stefan cel Mare si Sfant BoulevardMD 2004 Chisinau, Republic of Moldova. [3]Department of Biomedicine, University of Bergen, Bergen, Norway. [4]Department of Cell Biology, Yale School of Medicine, New Haven, CT, USA. [5]Université Paris Cité, Imagine Institute, Laboratory of Neurogenetics and Neuroinflammation, INSERM UMR 1163, Paris, France. [6]Hospices Civils de Lyon, Groupement Hospitalier Sud, Service d'Explorations Fonctionnelles Neurologiques, Lyon, France. [7]Department of Translational Genomics, Center for Genomic Medicine, King Faisal Specialist Hospital and Research Center, Riyadh, Saudi Arabia. [8]Department of Clinical and Movement Neurosciences, UCL Queen Square Institute of Neurology, London WC1N 3BG, UK. [9]Univ Rouen Normandie, Inserm U1245, CHU Rouen, Department of Genetics and CNRMAJ, F-76000 Rouen, France. [10]Department of Clinical and Experimental Epilepsy, UCL Queen Square Institute of Neurology, London WC1N 3BG, UK. [11]South Kazakhstan Medical Academy Shymkent, Shymkent 160019, Kazakhstan. [12]Instituto de Investigaciones en Medicina Traslacional (IIMT), CONICET-Universidad Austral, Av. Juan Domingo Perón 1500, B1629AHJ Pilar, Argentina. [13]Instituto de medicina genómica (IMeG), Hospital Universitario Austral, Universidad Austral, Av. Juan Domingo Perón 1500, B1629AHJ Pilar, Argentina. [14]Department of Human Genetics, Medical Research Institute, Alexandria University, Alexandria, Egypt. [15]Oxford NIHR Biomedical Research Centre, Wellcome Centre for Human Genetics, Oxford, United Kingdom. [16]Department of Clinical Science, University of Bergen, 5020 Bergen, Norway. [17]Institut de Recherche en Infectiologie de Montpellier, Université de Montpellier, CNRS, Montpellier, France. [18]Department of Radiology, Autism & Developmental Medicine Institute, Geisinger, Lewisburg, PA, USA. [19]Department of Radiology, Neuroradiology Section, Kingston Health Sciences Centre, Queen's University Faculty of Health Sciences, Kingston, Ontario, Canada. [20]Université Paris-Saclay, CEA, Centre National de Recherche en Génomique Humaine (CNRGH), 91057 Evry, France. [21]Neurology Department, Strasbourg University Hospital, Strasbourg, France. [22]Strasbourg Federation of Translational Medicine (FMTS), Strasbourg University, Strasbourg, France. [23]INSERM-U964; CNRS-UMR7104, University of Strasbourg, Illkirch-Graffenstaden, France. [24]Institute of Human Genetics, University Medical Center Göttingen, 37073 Göttingen, Germany. [25]Institute for Auditory Neuroscience and InnerEarLab, University Medical Center Göttingen, 37075 Göttingen, Germany. [26]Department of Neuroscience Center, King Faisal Specialist Hospital and Research Center, Riyadh, Saudi Arabia. [27]Paediatric Neurology, Faculty of Medicine, Dicle University, Diyarbakır, Turkey. [28]Centogene GmbH, Am Strande 7, 18055 Rostock, Germany. [29]Inonu University, Faculty of Medicine, Turgut Ozal Research Center, Department of Pediatrics, Division of Pediatric Neurology, Malatya, Turkey. [30]Dokuz Eylul University, School of Medicine, Department of Paediatric Neurology, Izmir, Turkey. [31]Children's Hospital of Eastern Ontario Research Institute and Division of Neurology, Department of Medicine, The Ottawa Hospital, Ottawa, Canada. [32]Brain and Mind Research Institute, University of Ottawa, Ottawa, Canada. [33]Department of Neuropediatrics and Muscle Disorders, Medical Center–University of Freiburg, Faculty of Medicine, Freiburg, Germany. [34]Department of Human Genetics, Sidra Medicine, Doha, Qatar. [35]College of Health and Life Sciences, Hamad Bin Khalifa University, Doha, Qatar. [36]Weill Cornell Medical College, Doha, Qatar. [37]Department of Information and Communications Engineering, University of Murcia, Campus Espinardo, 30100 Murcia, Spain. [38]Neurogenetics Laboratory, The National Hospital for Neurology and Neurosurgery, London WC1N 3BG, UK. [39]Department of Clinical Neurosciences, University of Cambridge, Cambridge, UK. [40]School of Biological Sciences, Department of Cell and Developmental Biology, University of California San Diego, La Jolla, CA, USA. [41]Medical Genetics Department, St George's University Hospitals, London SWI7 0RE, UK. [42]MRC Human Genetics Unit, Institute of Genetics and Cancer, University of Edinburgh, Edinburgh, UK. [43]Department of Anatomy and Cell Biology, College of Medicine, Alfaisal University, Riyadh, Saudi Arabia. [44]Department of Surgery, Haukeland University Hospital, Bergen, Norway. [46]These authors contributed equally: Viorica Chelban, Henriette Aksnes. [47]These authors jointly supervised this work: Thomas Arnesen, Henry Houlden. *A list of authors and their affiliations appears at the end of the paper. ✉e-mail: v.chelban@ucl.ac.uk; henriette.aksnes@uib.no; thomas.arnesen@uib.no; h.houlden@ucl.ac.uk

## SYNaPS Study Group

Rauan Kaiyrzhanov[1,11], Reza Maroofian [1], Stephanie Efthymiou [1], Vincenzo Salpietro [1], Henry Houlden[1] & Fowzan S. Alkuraya[45]

[45]King Faisal Specialist Hospital and Research Center, Riyadh, Saudi Arabia. A full list of members and their affiliations appears in the Supplementary Information.

