## [Peer Review File · Nature Communications]

Biallelic NAA60 Variants with Impaired N-terminal Acetylation Capacity Cause Autosomal Recessive Primary Familial Brain CalcificationsREVIEWER COMMENTS

Reviewer #1 (Remarks to the Author):

This research identified that biallelic variants of NAA60 gene were the novel genetic cause of PFBC from five families. And this study revealed that SLC20A2, the most common disease-causing gene for PFBC, may be the substrate of NAA60 in vitro. It may provide the novel molecular mechanism of PFBC. The genetic data were solid and convincing, while the functional results could not directly reflect the essence of disease.

Major comments:

1. Did the authors established the NAA60-KO mouse model and observe the calcification phenotype in NAA60-KO mice?
2. How the Golgi dysfunction caused by NAA60 loss-of-function should be discussed in detail.
3. It is interesting that all the five families carried homozygous variants. Were the F1 and F3 families consanguineous?
4. Whether the Pi level of the patients with biallelic NAA60 variants was higher than that of healthy controls?
5. To confirm the bona-fide substrates of NAA60, it is recommended to carry out the mass spectrum analysis, especially the endogenous data, which could be more objective. At least, you should examine the cellular expression pattern as the theoretic basis for NAA60/SLC20A2 pathway.
6. Our team have tested the SLC20A2 antibody as listed, but we found it could not distinguish the endogenous pit2 protein bands between WT and SLC20A2-KO mice. Please show the specificity validation of this SLC20A2 antibody (Ab191182).
7. According to the conclusion of previous study (PMID: 32393577), there is some interplay between the phosphate transporters like SLC20A2 and XPR1. So, how to explain your results which indicate only the uptake process was impaired?
8. The zebrafish model is not convincing. Did the authors analyze the calcification phenotype in the naa60 knockout or knockdown zebrafish?

Minor comments:

1. Table 1 could be more concise.
2. There are some grammatical errors, for example, "Similarly" but not "Similar" in line 261; "SLC20A2" but not "SLC20A" in line 294.

Reviewer #2 (Remarks to the Author):

The authors of the manuscript discovered a new disease gene (NAA60) linked to primary familial brain calcification (PFBC). This gene is the eighth disease gene discovered so far for PFBC, since recently another new gene, CMPK2, has been reported (Zhao M. et al, 2022 - Cell Discovery), which I suggest the authors to cite.

The findings reported in the present work, extend the genotype/phenotype associations in PFBC and are of importance since can contribute to improve the diagnostic rate of this disease, for which around 50% of cases remain genetically undiagnosed. In addition, this work deepens the knowledge on the impairment of brain Pi homeostasis, one of the three biological mechanisms known to be at the basis of PFBC pathogenesis, and highlights how post translational modifications, on proteins already known to be involved in the disease, could play a crucial role that needs to be furtherly investigated.

In this work, candidate pathogenic biallelic variants in NAA60 were found initially in one family through a combination of WES and homozygosity mapping analysis. Subsequently, after an extended screening on 180 cases from 83 families with PFBC and on two large genomics databases (Solve RD-Connect platform and 100,000 Genomes Project) the authors identified four additional PFBC families with candidate NAA60 variants. Although the genetic analysis has been properly performed through the use of WES, WGS and homozygosity mapping, I suggest to add a CNVs analysis on NGS data in order to potentially find new candidate NAA60 variants.

Genetic findings were exhaustively validated in vitro (performing experiments on HeLa, HAP1, RPE1, HEK293T and human fibroblasts cell lines) and in vivo (on zebrafish KO model). The authors described how mutated NAA60 proteins fail to associate with the Golgi and lack intrinsic enzymatic activity, potentially leading to reduced (Nt)-acetylation of several target proteins, including SLC20A2. Defects in Pi transport were described in both in vitro and in vivo models, suggesting that the impaired targeting or function of these non-Nt-acetylated NAA60 substrates may lead to impairment of phosphate and calcium phosphate homeostasis in the brains of patients with loss-of-function variants in NAA60. Data produced have been clearly presented and described in the figures and tables (including in supplementary files), statistical analysis have been properly performed and the experimental approach has been accurately described in details in the methods section.

Here are my comments regarding the text of the manuscript:

Line 113

In the abstract the authors stated: “we identify biallelic NAA60 variants in seven individuals from four families with autosomal recessive PFBC”. However, at line 318 and in Table 1, the authors wrote that variants in NAA60 were found in eight individuals from five unrelated families. Please correct the sentence in the abstract.

Line 134-135

The authors should add the references of the original article for each gene linked to the disease. In addition, a recent paper (Zhao M. et al – Cell Discovery) published on November 29th discovered a new gene (CMPK2) associated with PFBC, I suggest to add it in the list of genes linked to the disease.

Line 251

Although the paper published by Aksnes H. et al (2015), cited in the manuscript, reports that the lack of NAA60 by RNA silencing affects Golgi morphology in HeLa and CAL-62 cells, in figure S2c of the present manuscript the HAP1 NAA60 KO cells don't show altered morphology of Golgi. Could the authors explain this aspect?

Figure 2d

I suggest to use empty dots or at least modifying them with a certain degree of transparency in order to be able to visualize the boxplot behind the colored dots.

Figure S3b-c

The authors could add data obtained using bioinformatics tool, such as Dynamut2, for predicting the effect of missense variations on protein stability and dynamics.

Figure 3h, h', j, l', m', n'

Are the error bars missing in these graphs?

For figure 3j the J is present twice, is the second one a K?

Figure m and n

Could the dark/light legend be added at the top of the graph (as in figure 3L)?

Figure 5

I suggest to use the same color for the transmembrane protein (SLC20A2), on the left (Golgi apparatus) and on the right (Plasma membrane) of the figure, or at least writing on the protein its name, like authors did for NAA60.

Line 434

Genome-wide sequencing

At Line 180 the authors wrote that F4 was analyzed by WGS, but at line 436 in methods section the authors wrote that WES was performed in families 1-4. In addition, for family F1, F3 (HiSeq4000) and F2 (HiSeq2000) the sequencing platform used is indicated, but for family F4 and F5 not.

Did the authors performed CNV calling on WES and WGS in order to find large deletions/duplications in addition to single point variants and small INDELS?

If not, I suggest to perform this analysis using dedicated tools, new potential disease causing variants could be found in NAA60 gene.

References

Line 1020: reference 9 in the text refers to resorufin (line 297), but the paper of Taglia I. et al 2015 is a review in which resorufin is not mentioned.

Line 1057: (!!! INVALID CITATION !!! 39,40). Reference 21 is missing.

Reviewer #3 (Remarks to the Author):

This manuscript reports clinical, genetics and functional studies of a new culprit for primary Brain calcification.

Since 2010, 4 genes with autosomal dominant pattern (slc20a2, pdgfrb, pdgfra and xpr1) were reported and 3 following a recessive pattern (myorg, jam-2). Very recently, cmpk2 was also reported in a recessive pattern of inheritance.

This article reports a major finding. However, I would like to suggest some references to be added, one reporting the recently found gene, also with a recessive pattern. Other reference is about a case where homozygous pathogenic variants in slc20a2 was found in a patient with a severe clinical outcome and calcifications resembling a STORCH pattern. Also it would be interesting to report a recent case where digenic inheritance was found, with both pdgfrb and slc20a2 pathogenic variations. This will enrich the discussion since the new gene reported in this marvelous manuscript is also connected with an effect at slc20a2 gene. We can start seeing a much more complex scenario that might help to understand other conditions with brain calcification and recessive pattern as usp 18 and ISG15. We can also unveil how complex is the interaction of the genes linked to Brain calcification with alternative patterns of inheritance, to avoid analysis that could simplify the understanding of the genetic associations leading to Brain calcifications.

This will enrich their model and discussion.

References:

<https://onlinelibrary.wiley.com/doi/epdf/10.1002/mgg3.1670>

https://www.nature.com/articles/s41421-022-00475-2?fbclid=IwAR033U3uoCy30PhFa_vVPFvxVgs7APYimsaWs9MdTs1SAK5zcyS_JjnvXM

<https://link.springer.com/article/10.1007/s13760-022-02044-6>

Reviewer #4 (Remarks to the Author):

This study presents a compelling case that NAA60 mutations that prevent SLC20A2 N-acetylation lead to Primary Familial Brain Calcifications (PFBC). The genetics is persuasive, the biochemical work is nicely done, and the subcellular localization studies are convincing. The data supports a cogent story in which NAA60 acetylation of the N-terminal of SLC20A2 is required for SLC20A2 cell surface expression and phosphate transport. Without NAA60 activity, PFBC occurs, similar to SLC20A2 mutations. However, the zebrafish experiments are much less impressive: these experiments do not support the assertion that zebrafish crispants have a motor deficit, and therefore do not reinforce the link between NAA60 mutations and the various motor impairments in patients.

Critically, the zebrafish sudden light exposure experiment can not by itself, reveal a motor deficit. Changes in this assay may reflect differences in retinal signaling, deep brain photoreceptor signaling, central processing, behavioral state, or motor output.

Second, even if one was to accept that differences were due to motor effects alone, the data in 4l-n does not support that mutants have reduced motility. Mean distance / velocity appears lower in the dark for mutants, but following 30 sec after the light exposure, mutants and wildtypes appear very similar, and statistical analysis of the interval from 30 sec to 2 mins after illumination might even show that mutants have more slightly greater movement. It is not very surprising that 5 dpf zebrafish lack an overt motility phenotype, given that all except for patient 5 - who is an outlier in many respects - have clinical onset well after infant/neonatal stages.

Third, crispants have off target effects due to cutting at sites outside the target sequence that are not easily predicted. It is not sufficient to rely on the algorithms that predict whether a given guide may have

off-target cutting. Thus, the phenotypes reported here may not be due to mutations in NAA60 but in another gene. It is essential to replicate experiments (Pi, and behavior) using a second set of guide RNAs that target different sequences.

Fourth, please add information about the efficiency with which crispants produced bi-allelic cutting, as this varies highly between guide RNAs. Were all larvae used for Pi and behavior also evaluated for the extent of bi-allelic cutting, and if so, how. This is essential because even in good experiments the amount of cutting varies extensively.

Fifth, what are the controls? Especially for the behavioral experiments, it is not adequate to use wildtype uninjected larvae. These need to be sibling larvae, that were injected with cas9 and either no sgRNA or some kind of control sgRNA. Wildtype larvae from the same strain, but different clutches are not ok for behavioral experiments, especially with a subtle phenotype.

Other concerns:

Throughout, the figure legends should specify what error bars are: sd, sem, ci ?

Line 164. Confusing how F5 was discovered - patients appears to be part of 100000 genomes project but is presented here as a separate individual.

The in vitro experiment in Fig S1c-e using synthetic exon control constructs in HEK cells is consistent with splice abnormalities but not definitive. Is patient RNA available?

Patient 5 appears to have a distinct set of clinical manifestations raising the possibility that pathology is due to different mutations in this individual that the authors say they tried to discover but could not find. Also from Fig S3A that residue appears less well conserved than neighbors. What additional genetic analyses were performed in this patient? Are any potentially pathogenic hemizygous mutations present?

While NAA60 mutations 1/2/3 all clearly alter the distribution to the cytosol/nucleus, the comparison of wildtype NAA60 with GM130 localization is intriguing. Most of the distribution would best be described as Golgi-adjacent rather than actually co-localizing. Could the authors please comment on what this might mean.

The morphology of the Golgi in Fig S2C is hard to determine (for NAA60), as the images are low resolution and saturated.

Fig. 2d lacks statistical significance indicators mentioned in the legend. Also, error bars hidden by points.

Fig. S4 legend states "Bottom: Luxol fast blue for visualization of myelin" but the panel is labeled 'cresyl violet'

The experiment in Fig. 3j should present data for each of the four groups tested, along with the statistical analysis - I think the bar chart just shows the the means. What does it mean in the methods for Pi in zebrafish that the statistics used "t-test, Welch's t-test" - just that the Welch's test was used? Or two different tests?

Schematic of behavioral test in 4k is mislabeled as 4j.

The figure legend for 4l,m,n lacks details needed to interpret the data. I initially assumed that mean distance/velocity/acceleration in 'l', 'm', 'n' were taken from the boxed areas (which is missing in 4l). But this does not make sense based on my estimation of what those numbers would be. In any case, because motility changes constantly during the first 30 s after illumination, it does not make sense to average numbers in that interval. Matched points should be statistically compared. No statistical comparison is presented for acceleration so I assume that it is not significant and the statement in the text that it is lower (line 315) should be removed. Similarly without a change in acceleration after the light-on, there is no data to support the statement on line 365 that zebrafish have a reduced light-response: the increase from baseline looks very similar.

Methods section describing zebrafish experiments lacks important details:

1. The crispant experiment refers for methods to reference 58. Reference 58 also contains no information about how crispants were created. Please include information about where reagents (cas9, sgRNA) were sourced from, prepared and injection concentrations - it is not possible without this to evaluate the likelihood of bi-allelic mutations.
2. Provide allele number and citations for all transgenic lines used.
3. Ethovision assay: light measurements during 'dark' and after illumination.

Point-by-point response to reviewer comments

We would like to thank all the reviewers for their thorough review of our work, constructive criticism, and useful comments, which we believe have helped us to improve our manuscript. We are pleased to report that during the review process, we have identified additional families and patients and the current cohort reported in this manuscript represents 10 patients from 7 unrelated families from different ethnic backgrounds. After evaluating the comments on the zebrafish model overall, we have decided to remove it from the manuscript as these results are not central to the findings of the study. We believe that the addition of new cases from unrelated families together with the functional work present a compelling case that NAA60 variants lead to Primary Familial Brain Calcifications through impaired Nt-acetylation.

Please find answers to the specific points raised by the reviewers below.

Reviewer #1 (Remarks to the Author):

This research identified that biallelic variants of NAA60 gene were the novel genetic cause of PFBC from five families. And this study revealed that SLC20A2, the most common disease-causing gene for PFBC, may be the substrate of NAA60 in vitro. It may provide the novel molecular mechanism of PFBC. The genetic data were solid and convincing, while the functional results could not directly reflect the essence of disease.

Major comments:

1. Did the authors established the NAA60-KO mouse model and observe the calcification phenotype in NAA60-KO mice?

- No, we have not established a NAA60-KO mouse model for this study. *This is the team's intention for further work into the condition at a later stage.*

2. How the Golgi dysfunction caused by NAA60 loss-of-function should be discussed in detail.

- We thank the reviewer for pointing this out. As similar comments were also made by other reviewers we have improved and expanded this experiment and the accompanying text has been updated. Previous Figure S2 has been updated to a main figure (now Fig. 4) and the accompanying text now reads: "Previously, it was shown that knockdown of NAA60 in HeLa cells results in a disruption of the Golgi ribbon architecture¹⁵. To determine whether the NAA60 frameshift variants reproduce this phenotype, we performed immunofluorescence microscopy of primary dermal fibroblasts from NAA60^{-/-} cases from F1. However, the Golgi structure appeared intact in all cells from both affected and unaffected individuals (Fig. 4b). We reasoned that this discrepancy with previous findings could be due to cell type differences or that it could be explained by differential effects of knockdown vs knockout. We therefore investigated a CRISPR/Cas9-generated NAA60 knockout cell line harbouring a frameshift at a position close to that of NAA60-PFBC in Families 1, 2 and 3 (Fig. 4c). Of note, these HAP1 NAA60^{-/-} cells also showed an intact Golgi apparatus (Fig. 4d), where NAA80 knockout was used as a positive control for Golgi fragmentation in the HAP1 cell line²². We next also extended this investigation by performing the same shRNA knockdown experiment as previously done in HeLa cells, but now in dermal fibroblasts from healthy controls. Here, we observed that NAA60-shRNA-transfected dermal fibroblast controls displayed fragmented Golgi (Figure 4e), similar to the previous work in HeLa cells¹⁵. Thus, we found support for the

idea that Golgi fragmentation occurs in RNA silencing experiments, but not in cells with permanent NAA60 absence as a result of an early frameshift in the genomic DNA.”

3. It is interesting that all the five families carried homozygous variants. Were the F1 and F3 families consanguineous?

- Family 1 was not known to be consanguineous. However, we cannot exclude a degree of distant relatedness as the family is from small rural area from the UK. We performed additional analysis using Peddy and Automap to assess consanguinity. The kinship coefficient between the siblings, calculated using Peddy, did not exceed the expected 0.5 value. Comparing the samples in Automap did not reveal a large number of shared runs of homozygosity to support close consanguinity. We added this information in the manuscript. Family 3 was consanguineous. This information is added in Table 1.

4. Whether the P_i level of the patients with biallelic NAA60 variants was higher than that of healthy controls?

- Patient CSF was not available. We assessed P_i levels for family 1 (in cultured dermal fibroblasts) and family 2 (peripheral blood mononuclear cells). This information is indicated in the main manuscript and in Figure 5h-i: “Medium from NAA60 F1 cases contained significantly higher P_i levels than medium from age-matched controls and an SLC20A2 missense mutant case associated with impaired function (Fig. 5h). We assessed the cell capacity for P_i depletion and found that there was a lower expenditure of P_i in cell cultures from NAA60 mutant cases (7.9%) compared to SLC20A2 mutant cases (36.1%) and controls (43.3%) (Fig. 5h’). In addition, investigation of fresh peripheral blood mononuclear cells from an affected patient in F2 also suggested similarly effect on P_i homeostasis (Fig. 5i-i’).”

5. To confirm the bona-fide substrates of NAA60, it is recommended to carry out the mass spectrum analysis, especially the endogenous data, which could be more objective.

- We agree with the reviewer that a global N-terminomic mass spectrometry analysis could potentially shed further light on the disease mechanisms. However, this analysis extends beyond the scope of this study. Furthermore, even state-of-the-art N-terminal acetylation mass spectrometry analysis detects only a minor fraction of the proteome due to protein abundance and N-terminal protein sequence. However, we have inspected our previous global dataset of endogenous membrane proteins (Aksnes et al., Cell Reports, 2015, PMID: 25732826) and unfortunately, the seven endogenous PFBC proteins were not detected in such setups (including CMPK2 which we have now also checked). This strongly suggest that the N-terminal acetylation status of these PFBC proteins is not easily analyzed at the endogenous level by mass spectrometry. In addition, we specifically attempted to assess the N-terminal acetylation status of SLC20A2 by mass spectrometry using overexpressed and immunoprecipitated SLC20A2, but the SLC20A2 N-terminus could unfortunately not be detected.

At least, you should examine the cellular expression pattern as the theoretic basis for NAA60/SLC20A2 pathway.

-We have added a new analysis based on your valuable suggestion and performed NAA60/SLC20A2 co-expression networks analysis. Full details of the results are presented in the revised manuscript but overall, this additional analysis shows: “Overall, these findings suggest that while the expression pathways of SLC20A2 and NAA60 are distinct, they are closely related especially in the nucleus accumbens. Furthermore, this data places NAA60 within a network of genes involved in morphological brain structure and

neurodevelopment in humans, particularly within the neurons from the caudate and nucleus accumbens of the basal ganglia, in keeping with the human NAA60-related phenotype.

6. Our team have tested the SLC20A2 antibody as listed, but we found it could not distinguish the endogenous pit2 protein bands between WT and SLC20A2-KO mice. Please show the specificity validation of this SLC20A2 antibody (Ab191182).

- We would like inquire whether this information is publicly available. We have reviewed the literature and were not able to identify any SLC20A2/PIT2 antibody that is KO-validated. The specifics of the antibody used here are: Anti-SLC20A2/PIT2 antibody (ab191182) Lot: GR201747-5. This work was performed in the Rothman laboratory some years back, likely with another batch/lot than what is currently available the market. We have, however, noticed that the immunogen used to generate this antibody maps to a region of SLC20A2 that has some sequence match with SLC20A1 ("Immunogen Synthetic peptide within Human SLC20A2/PIT2 aa 300-400"). It can therefore not be excluded that this antibody could also have reactivity towards SLC20A1 which has a very similar molecular weight. We have added a comment about this in the Methods section under *Surface biotinylation* where the specifics of the antibody are listed.

7. According to the conclusion of previous study (PMID: 32393577), there is some interplay between the phosphate transporters like SLC20A2 and XPR1. So, how to explain your results which indicate only the uptake process was impaired?

- Thank you for this suggestion. We have added clarification on this point in the discussion: "Recently, it was unveiled an interplay between SLC20A2 and XPR1 with functional transport activities that appeared to control cellular phosphate homeostasis, although only a defect in XPR1 led to increased global phosphate concentrations, which is quite in contrast to defective PFBC SLC20A2 variants, which affected neither phosphate and ATP levels nor phosphate efflux. This is consistent with our results, with variants in NAA60 affecting the SLC20A2 function with similar effect on cellular P_i homeostasis."

8. The zebrafish model is not convincing. Did the authors analyse the calcification phenotype in the naa60 knockout or knockdown zebrafish?

- We have decided to remove the zebra fish from the manuscript.

Minor comments:

1. Table 1 could be more concise.

Thank you for the suggestion. We have shortened the Table 1. We have also removed the predictions and ACMG classification of the variants from this table and have created a new table with more detailed information using in silico tools for further predicting the effects (Supplementary table 2).

2. There are some grammatical errors, for example, "Similarly" but not "Similar" in line 261; "SLC20A2" but not "SLC20A" in line 294.

- These have been corrected, thank you.

Reviewer #2 (Remarks to the Author):

The authors of the manuscript discovered a new disease gene (NAA60) linked to primary familial brain calcification (PFBC). This gene is the eighth disease gene discovered so far for PFBC, since recently another new gene, CMPK2, has been reported (Zhao M. et al, 2022 - Cell Discovery), which I suggest the authors to cite.

- The paper was now cited as a gene that presents with brain calcifications, thank you for your suggestion. However, we did not include it as PFBC-related gene. Loss of CMPK2 indeed induces brain calcification but also mitochondrial disorders. Mitochondriopathies are known to be associated with brain calcification and the identified gene variants are not considered as PFBC genes. The same is true for interferonopathies like Aicardi-Goutière syndrome which causes auto-inflammation and brain calcification for which the associated genes (TREX1, RNase H2, SAMHD1, etc) are not PFBC genes. In their paper, Zhao et al did not state that CMPK2 is a PFBC-related gene: "CMPK2 expression was also found to be regulated upon the stimulation with type I interferon (40,41), suggesting possible interplay between CMPK2 and interferon responses. As interferon stimulation has been previously shown to promote the progression of brain calcification (42), investigation of any involvement of CMPK2 in the interferon pathway in human or mouse brains may help expand our understanding of how CMPK2 deficiency causes brain calcification". In our collective opinion, we consider it too early to say that CMPK2 is a bonafide PFBC gene, however we cite it in the manuscript as a gene that presents with brain calcifications.

The findings reported in the present work, extend the genotype/phenotype associations in PFBC and are of importance since can contribute to improve the diagnostic rate of this disease, for which around 50% of cases remain genetically undiagnosed. In addition, this work deepens the knowledge on the impairment of brain Pi homeostasis, one of the three biological mechanisms known to be at the basis of PFBC pathogenesis, and highlights how post translational modifications, on proteins already known to be involved in the disease, could play a crucial role that needs to be furtherly investigated.

In this work, candidate pathogenic biallelic variants in NAA60 were found initially in one family through a combination of WES and homozygosity mapping analysis. Subsequently, after an extended screening on 180 cases from 83 families with PFBC and on two large genomics databases (Solve RD-Connect platform and 100,000 Genomes Project) the authors identified four additional PFBC families with candidate NAA60 variants. Although the genetic analysis has been properly performed through the use of WES, WGS and homozygosity mapping, I suggest adding a CNVs analysis on NGS data in order to potentially find new candidate NAA60 variants.

- Thank you for suggesting this, we have performed CNV analysis already, but we had no CNV variants in our cohorts. We have now included this information in methods. "Structural variants detected in 33,924 families with WGS data were prioritized using a MySQL database of variants called by Manta and CANVAS, as described⁵⁰."

Genetic findings were exhaustively validated in vitro (performing experiments on HeLa, HAP1, RPE1, HEK293T and human fibroblasts cell lines) and in vivo (on zebrafish KO model). The authors described how mutated NAA60 proteins fail to associate with the Golgi and lack intrinsic enzymatic activity, potentially leading to reduced (Nt)-acetylation of several target proteins, including SLC20A2. Defects in Pi transport were described in both in vitro and in vivo models, suggesting that the impaired targeting or function of these non-Nt-acetylated NAA60 substrates may lead to impairment of phosphate and calcium phosphate homeostasis in the brains of patients with loss-of-function variants in NAA60. Data produced have been clearly presented and described in the figures and tables (including in supplementary files), statistical analysis have been properly performed and the experimental approach has been accurately described in detail in the methods section.

Here are my comments regarding the text of the manuscript:

Line 113

In the abstract the authors stated: “we identify biallelic NAA60 variants in seven individuals from four families with autosomal recessive PFBC”. However, at line 318 and in Table 1, the authors wrote that variants in NAA60 were found in eight individuals from five unrelated families. Please correct the sentence in the abstract.

- This has been corrected through the manuscript, thank you. We now include ten individuals from seven families.

Line 134-135

The authors should add the references of the original article for each gene linked to the disease. In addition, a recent paper (Zhao M. et al – Cell Discovery) published on November 29th discovered a new gene (CMPK2) associated with PFBC, I suggest adding it in the list of genes linked to the disease.

- Thank you for this suggestion, we have added the references for the original articles for each PFBC gene and the *CMPK2* gene was cited as a causative gene for FBC.

Line 251

Although the paper published by Aksnes H. et al (2015), cited in the manuscript, reports that the lack of NAA60 by RNA silencing affects Golgi morphology in HeLa and CAL-62 cells, in figure S2c of the present manuscript the HAP1 NAA60 KO cells don't show altered morphology of Golgi. Could the authors explain this aspect?

- Indeed, we have found a discrepancy between the previous data on HeLa cells depleted for NAA60 with siRNA/shRNA approaches and the dermal fibroblast from NAA60 knockout patients. We reasoned that this could be due to cell type differences or that it could be explained by differential effects of knockdown vs knockout. We thus investigated the HAP1 knockout model and found support for the latter. In the revision process we extended this analysis by performing the same shRNA knockdown in dermal fibroblasts from healthy controls. Here, we observed that shRNA-transfected dermal fibroblast controls displayed fragmented Golgi, similar as the previous work in HeLa cells (new Figure 4e). Thus, again further confirming that Golgi fragmentation occurs in depletion experiments, but not in cells with permanent NAA60 absence as a result of an early frameshift. This was clarified and the new experiments were added in the manuscript: “We next extended this investigation by performing the same shRNA knockdown experiment as previously done in HeLa cells, but now in dermal fibroblasts from healthy controls. Here, we observed that *NAA60*-shRNA-transfected dermal fibroblast controls displayed fragmented Golgi (Fig. 4e), similar as the previous work in HeLa cells¹⁵. Thus, we found support for the idea that Golgi fragmentation occurs in RNA silencing experiments, but not in cells with permanent NAA60 absence due to early frameshift in the genomic DNA”.

Figure 2d

I suggest to use empty dots or at least modifying them with a certain degree of transparency in order to be able to visualize the boxplot behind the colored dots.

- We thank the reviewer for pointing this out. The data-visualization has been improved as suggested and the information on statistical test has also been added in the plot and in the legend. This is now Figure 3 as new information was added during revision.

Figure S3b-c

The authors could add data obtained using bioinformatics tool, such as Dynamut2, for predicting the effect of missense variations on protein stability and dynamics.

- We thank the reviewer for this suggestion. We have included this analysis for all missense mutants (including the two additional families F6 and F7) in a new Figure S4 and in a new figure panel (Fig. 2) and commented in the accompanying text.

Figure 3h, h', j, l', m', n'
Are the error bars missing in these graphs?

- Thank you, error bars have been included in all bar charts.

For figure 3j the J is present twice, is the second one a K?

- Yes, thank you for pointing this out. These data have been removed from the manuscript after the input from another reviewer.

Figure m and n
Could the dark/light legend be added at the top of the graph (as in figure 3L)?

- We thank the reviewer for this suggestion to improve the data presentation. These data have in any case been removed from the manuscript after the input from another reviewer.

Figure 5
I suggest using the same color for the transmembrane protein (SLC20A2), on the left (Golgi apparatus) and on the right (Plasma membrane) of the figure, or at least writing on the protein its name, like authors did for NAA60.

- We agree that using orange colour also on the left side of the figure would highlight the suggested role of SLC20A2. However, we here wanted to be careful not to exclude the option that there could be other unidentified NAA60 substrates involved. We therefore used the grey colour on the left to indicate a general membrane protein that could be SLC20A2. We have now updated the text in the figure to better convey this.

Line 434
Genome-wide sequencing

At Line 180 the authors wrote that F4 was analyzed by WGS, but at line 436 in methods section the authors wrote that WES was performed in families 1-4.

- This was corrected in the methods. "Whole-exome sequencing (WES) was performed in all families except Family 5, which had whole genome sequencing (WGS) performed»

In addition, for family F1, F3 (HiSeq4000) and F2 (HiSeq2000) the sequencing platform used is indicated, but for family F4 and F5 not.

- Thank you, we have now included the information for all families.

Did the authors performed CNV calling on WES and WGS in order to find large deletions/duplications in addition to single point variants and small INDELS?
If not, I suggest to perform this analysis using dedicated tools, new potential disease causing variants could be found in NAA60 gene.

- Thank you for suggesting this, we have performed CNV analysis already, but we had no CNV variants in our cohorts. We have now included this information in methods. "Structural variants detected in 33,924 families with WGS data were prioritised using a MySQL database of variants called by Manta and CANVAS, as described⁵⁰."

References

Line 1020: reference 9 in the text refers to resorufin (line 297), but the paper of Taglia I. et al 2015 is a review in which resorufin is not mentioned.

- We have clarified the sentence and the reference indicating that the reference describes the method and the kit used for that method.

Line 1057: (!!! INVALID CITATION !!! 39,40). Reference 21 is missing.

- Thank you, this error has been fixed now.

Reviewer #3 (Remarks to the Author):

This manuscript reports clinical, genetics and functional studies of a new culprit for primary Brain calcification.

Since 2010, 4 genes with autosomal dominant pattern (slc20a2, pdgfrb, pdgfb and xpr1) were reported and 3 following a recessive pattern (myorg, jam-2). Very recently, cmpk2 was also reported in a recessive pattern of inheritance.

This article reports a major finding. However, I would like to suggest some references to be added, one reporting the recently found gene, also with a recessive pattern.

- Thank you, this was added now.

Other references is about a case Where homozygous pathogenic variants in slc20a2 was found in a patient with a severe clinical outcome and calcifications resembling a STORCH pattern.

Also it would be interesting to report a recent case where digenic inheritance was found, with both pdgfrb and slc20a2 pathogenic variations. This will enrich the discussion since the new gene reported in this marvellous manuscript is also connected with na effect at slc20a2 gene.

- Thank you, this was added now.

We can start seeing a Much more complex scenario that might help to understand other conditions with brain calcification and recessive pattern as usp 18 and ISG15. We can also unvail How complex os the interaction of the genes linked to Brain calcification with alternative patterns of inheritance, to avoid analysis that could simplify the understanding of the genetic associations leading to Brain calcifications. This Will enrich their model and discussion.

References:

<https://onlinelibrary.wiley.com/doi/epdf/10.1002/mgg3.1670>

[https://www.nature.com/articles/s41421-022-00475-](https://www.nature.com/articles/s41421-022-00475-2?fbclid=IwAR033U3uoCy30PhFa_vVPFvxVgs7APYimsaWs9MdTs1SAK5zcyS_JjnvXM)

[2?fbclid=IwAR033U3uoCy30PhFa_vVPFvxVgs7APYimsaWs9MdTs1SAK5zcyS_JjnvXM](https://www.nature.com/articles/s41421-022-00475-2?fbclid=IwAR033U3uoCy30PhFa_vVPFvxVgs7APYimsaWs9MdTs1SAK5zcyS_JjnvXM)

<https://link.springer.com/article/10.1007/s13760-022-02044-6>

- Thank you for these suggestions. We have included them all and added them to the discussion: "Moreover, a complex PFBC genotype-phenotype spectrum emerges as cases that involve dose-dependent variants as well as digenic PFBC-related variants have been associated severe clinical and neuroimaging spectrum. SLC20A2 heterozygous mutations are associated with the adult-onset phenotype, while homozygous SLC20A2 variants resulted in severe phenotype including growth retardation, microcephaly, and convulsion, similar to some of the NAA60 variants reported here."

Reviewer #4 (Remarks to the Author):

This study presents a compelling case that NAA60 mutations that prevent SLC20A2 N-acetylation lead to Primary Familial Brain Calcifications (PFBC). The genetics is persuasive, the biochemical work is nicely done, and the subcellular localization studies are convincing. The data supports a cogent story in which NAA60 acetylation of the N-terminal of SLC20A2 is required for SLC20A2 cell surface expression and phosphate transport. Without NAA60 activity, PFBC occurs, similar to SLC20A2 mutations.

However, the zebrafish experiments are much less impressive: these experiments do not support the assertion that zebrafish crispants have a motor deficit, and therefore do not reinforce the link between NAA60 mutations and the various motor impairments in patients.

Critically, the zebrafish sudden light exposure experiment can not by itself, reveal a motor deficit. Changes in this assay may reflect differences in retinal signaling, deep brain photoreceptor signaling, central processing, behavioral state, or motor output.

Second, even if one was to accept that differences were due to motor effects alone, the data in 4l-n does not support that mutants have reduced motility. Mean distance / velocity appears lower in the dark for mutants, but following 30 sec after the light exposure, mutants and wildtypes appear very similar, and statistical analysis of the interval from 30 sec to 2 mins after illumination might even show that mutants have more slightly greater movement. It is not very surprising that 5 dpf zebrafish lack an overt motility phenotype, given that all except for patient 5 - who is an outlier in many respects - have clinical onset well after infant/neonatal stages.

Third, crispants have off target effects due to cutting at sites outside the target sequence that are not easily predicted. It is not sufficient to rely on the algorithms that predict whether a given guide may have off-target cutting. Thus, the phenotypes reported here may not be due to mutations in NAA60 but in another gene. It is essential to replicate experiments (Pi, and behavior) using a second set of guide RNAs that target different sequences.

Fourth, please add information about the efficiency with which crispants produced bi-allelic cutting, as this varies highly between guide RNAs. Were all larvae used for Pi and behavior also evaluated for the extent of bi-allelic cutting, and if so, how. This is essential because even in good experiments the amount of cutting varies extensively.

Fifth, what are the controls? Especially for the behavioral experiments, it is not adequate to use wildtype uninjected larvae. These need to be sibling larvae, that were injected with cas9 and either no sgRNA or some kind of control sgRNA. Wildtype larvae from the same strain, but different clutches are not ok for behavioral experiments, especially with a subtle phenotype.

- After evaluating the comments on the zebrafish model overall, we have decided to remove it from the manuscript. We believe that the addition of new cases from unrelated families together with the functional work present a compelling case that NAA60 variants lead to Primary Familial Brain Calcifications through impaired Nt-acetylation.

Other concerns:

Throughout, the figure legends should specify what error bars are: sd, sem, ci ?

- Thank you. We have updated the figures and legends to specify this information.

Line 164. Confusing how F5 was discovered - patients appears to be part of 100000 genomes project but is presented here as a separate individual.

- Thank you. We have now provided detailed clarification which cohorts were screened and how each family was discovered.

The in vitro experiment in Fig S1c-e using synthetic exon control constructs in HEK cells is consistent with splice abnormalities but not definitive. Is patient RNA available?

- Unfortunately, the patient RNA is not available for this work.

*Patient 5 appears to have a distinct set of clinical manifestations raising the possibility that pathology is due to different mutations in this individual that the authors say they tried to discover but could not find. Also from Fig S3A that residue appears less well conserved than neighbours. **What additional genetic analyses were performed in this patient? Are any potentially pathogenic hemizygous mutations present?***

- "The proband in F5 did not show any calcifications on CT at the age of 12 years. The clinical presentation appeared to be more complex than the other patients that we ascertained. Although the complex clinical presentation raises the possibility of a co-existing diagnosis, exome sequencing did not reveal any additional variant that would explain the other phenotypic features. Given the NAA60 variant leading to abnormal NAA60 enzymatic activity and as at the time of the CT scan, this case was likely below the age of onset of when calcifications become visible on scans, we have included this variant here and follow-up of this patient will be important to conclude the existence of abnormal brain calcification during aging."

While NAA60 mutations 1/2/3 all clearly alter the distribution to the cytosol/nucleus, the comparison of wildtype NAA60 with GM130 localization is intriguing. Most of the distribution would best be described as Golgi-adjacent rather than actually co-localizing.

Could the authors please comment on what this might mean.

- We agree with the reviewer that our microscopy data reveal a more precise subcellular localization of NAA60 compared to previous work using older instruments (Aksnes et al Cell Reports 2015) as it shows that NAA60 and GM130 do not perfectly co-localize. We have added a brief comment about this. Noteworthy, GM130 is a Golgi protein with its main localization to the cis-Golgi. NAA60 appearing adjacent to it likely means that NAA60 prefers another sub-compartment of the Golgi (medial or trans). It could be possible to map this by using antibody markers towards Golgi proteins with distinct sub-Golgi localizations. However, it is not needed for the purpose of stating that the WT localization is intact or lost in the various mutants, which is what we aimed for here. We clarified this in the manuscript: "*The microscopy in Fig. 2b indicated that NAA60 did not perfectly co-localize with the Golgi marker GM130. Therefore, we performed structured illumination microscopy and co-localization analysis, which revealed that FLAG-NAA60-WT was co-distributed with the cis-Golgi marker GM130, as it localized to structures closely associated to, but not completely overlapping with the structures stained by GM130, (Fig. 4a, and Video S2). This likely means that NAA60 prefers another sub-compartment of the Golgi. Nonetheless, we found that GM130 is a valid marker for investigating intact versus disturbed Golgi localization of NAA60 and for assessing Golgi morphology.*"

The morphology of the Golgi in Fig S2C is hard to determine (for NAA60), as the images are low resolution and saturated.

- We agree with the comment. We have redone the imaging and renewed the figure (now this data is in Fig 4d) to provide microscopy data of the same standard as the remainder of the manuscript. We also extended the experiment to include shRNA-transfected control dermal fibroblasts as explained in response to above reviewer.

Fig. 2d lacks statistical significance indicators mentioned in the legend. Also, error bars hidden by points.

- We thank the reviewer for pointing this out. The figure and legend has been updated accordingly.

Fig. S4 legend states "Bottom: Luxol fast blue for visualization of myelin" but the panel is labeled 'cresyl violet'

The experiment in Fig. 3j should present data for each of the four groups tested, along with the statistical analysis - I think the bar chart just shows the the means. What does it mean in the methods for Pi in zebrafish that the statistics used "t-test, Welch's t-test" - just that the Welch's test was used? Or two different tests?

- This data has been removed from the manuscript.

Schematic of behavioral test in 4k is mislabeled as 4j.

- Yes, thank you for pointing this out. These data have been removed from the manuscript.

The figure legend for 4l,m,n lacks details needed to interpret the data. I initially assumed that mean distance/velocity/acceleration in l', m', n' were taken from the boxed areas (which is missing in 4l). But this does not make sense based on my estimation of what those numbers would be. In any case, because motility changes constantly during the first 30 s after illumination, it does not make sense to average numbers in that interval. Matched points should be statistically compared. No statistical comparison is presented for acceleration so I assume that it is not significant and the statement in the text that it is lower (line 315) should be removed. Similarly without a change in acceleration after the light-on, there is no data to support the statement on line 365 that zebrafish have a reduced light-response: the increase from baseline looks very similar.

- This data has been removed from the manuscript.

Methods section describing zebrafish experiments lacks important details:

1. The crispant experiment refers for methods to reference 58. Reference 58 also contains no information about how crispants were created. Please include information about where reagents (cas9, sgRNA) were sourced from, prepared and injection concentrations - it is not possible without this to evaluate the likelihood of bi-allelic mutations.

2. Provide allele number and citations for all transgenic lines used.

3. Ethovision assay: light measurements during 'dark' and after illumination.

- The zebra fish model has been removed from the manuscript.

REVIEWER COMMENTS

Reviewer #1 (Remarks to the Author):

Most of my confusions were addressed. But there were still several issues should be answered and modified.

1. The brain calcification of the proband of Family 6 is not typical, maybe due to the age. But it is suggested to remove this family from this work. Other families were enough.
2. The revised manuscript did not answer the question about the cell types of SLC20A2 and NAA60. Do they locate in the same cell type? This is the basis for any mechanism exploration. At least you could apply immunofluorescence assay or ISH assay to detect whether they were co-localized in the same cell type.
3. The authors are able to perform the Co-IP assay for SLC20A2 and NAA60, at least in vitro.

Reviewer #2 (Remarks to the Author):

The authors of the manuscript discovered a new disease gene (NAA60) linked to primary familial brain calcification (PFBC). These findings extend the genotype/phenotype associations in PFBC and are of importance since can contribute to improve the diagnostic rate of this disease, for which around 50% of cases remain genetically undiagnosed. In addition, this work deepens the knowledge on the impairment of brain Pi homeostasis, one of the three biological mechanisms known to be at the basis of PFBC pathogenesis, and highlights how post translational modifications, on proteins already known to be involved in the disease, could play a crucial role.

The manuscript has been significantly improved, in general agreement with the recommendations, authors have responded to my questions/comments in a satisfactory manner. The manuscript is ready for being accepted.

Reviewer #3 (Remarks to the Author):

Dear Authors,

Congratulations for the contribution and the update of this new version.

I believe that the results are relevante and there is a sound methodology behind the analysis.

There is enough details to accept the final conclusions and inferences.

Best regards,

João Oliveira

Federal University of Pernambuco

Reviewer #4 (Remarks to the Author):

The authors have responded to each of my concerns thoroughly, correcting minor errors and adjusting interpretations in the text. The higher quality images are much more satisfactory. It was unfortunate that the zebrafish experiments needed to be removed, but in this case I think the patient mutations and in vitro studies are sufficient to link NAA60 variants to brain calcifications.

We would like to thank all the reviewers for their thorough review of our work, constructive criticism and useful comments, which we believe have helped us to improve our manuscript. Please find the answers to the specific points raised by the reviewers and editorial team below.

Reviewer #1 (Remarks to the Author):

Most of my confusions were addressed. But there were still several issues should be answered and modified.

1. The brain calcification of the proband of Family 6 is not typical, maybe due to the age. But it is suggested to remove this family from this work. Other families were enough.

In the discussion section, we describe the pattern of calcifications in this case as follows: “patient from Family 6 exhibited very small calcifications at the age of 12, including two calcified spots in the periventricular white matter and in the left putamen, next to the external capsule.” The putamen, part of the basal ganglia, is typically affected in this condition. Although the calcifications have not yet spread to the contralateral side at the time of scanning, nevertheless, we consider it to be relevant to the condition as the young age at the time of scan would explain the limited nature of the calcification. Furthermore, the genetics and protein prediction data for NAA60-F6 is at the same level as the remaining missense variants. However, we agree that the imaging in this patient should be followed-up over time, therefore we have included in the discussion section the following statement: “The CT scans of the patients reported here are some of the youngest FPBC scans reported in literature, therefore young cases from F5 and F6 should be followed up to determine the pattern of progression of calcification on neuroimaging over time.”

2. The revised manuscript did not answer the question about the cell types of SLC20A2 and NAA60. Do they locate in the same cell type? This is the basis for any mechanism exploration. At least you could apply immunofluorescence assay or ISH assay to detect whether they were co-localized in the same cell type.

Thank you for the suggestion. We have addressed in the manuscript the co-expression of NAA60 and SLC20A2 as follows: “According to GTEx, both NAA60 and SLC20A are expressed in all tested human tissues (revised Figure S5a). Further, according to Human Protein Atlas, both these genes are also expressed in all tested single cell types from different human tissues confirming that they may act in the same cell”. This has now also been mentioned in the revised Results. *SLC20A2* and *NAA60* are also expressed in the same brain regions (Figure S5 and accompanying Results text). Please notice the co-expression network performed with Weighted Gene Co-expression Network Analyses (WGCNA), showing that NAA60 and SLC20A2 are closely related especially in the nucleus accumbens (Figure S5d-f). Furthermore, please notice that the healthy control dermal fibroblast cells express both proteins endogenously, Figure 3d (NAA60) and Figure 5e (SLC20A2). Immunohistochemical or immunocytochemical analysis would not provide significant further evidence for cell type co-expression, and this would also depend on immunofluorescence-compatible specific antibodies for both proteins which at present unfortunately are not available.

3. The authors are able to perform the Co-IP assay for SLC20A2 and NAA60, at least in vitro.

NAA60 is an N-terminal acetyltransferase which performs a transient catalytic reaction in transferring an acetyl moiety from acetyl coenzyme A to an amino group of a substrate protein. In contrast to some lysine acetyltransferases (KATs) where there may be relatively strong and stable cellular interactions between the KAT enzyme and the substrate enabling detection of enzyme-substrate complexes by co-immunoprecipitation analysis, N-terminal acetyltransferases (NATs) are normally not able to co-immunoprecipitate their substrates. This is seen both for co-translational NATs (Arnesen T *et al.*, *Biochem J*, 2005 386: 433–443) as well as post-translational NATs (Ree R *et al.*, *J Biol Chem*, 2020 295(49):16713-16731; unpublished data). Furthermore, NAA60 is mainly localized to the Golgi apparatus where it is believed to N-terminally acetylate transmembrane proteins *en route* to other destinations (Aksnes H *et al.*, *Cell Rep*, 2015 10:1362-74). Thus, NAA60 is not expected to coIP its substrates, and most likely SLC20A2 would not coIP with NAA60 even if it is a true *in vivo* substrate of NAA60. The transient interaction between NAA60 and SLC20A2 *in vitro* is represented by the ability of NAA60 to transfer an acetyl group from acetyl coenzyme A to the specific N-terminal sequence of SLC20A2 (Figure 5a-c). SLC20A2 thus represents one possible NAA60 substrate involved in the mechanism of how impaired NAA60 mediated N-terminal acetylation may cause PFBC. This uncertainty is stated in the Results and Discussion ('*In this work, we present pathogenic NAA60 variants as causative for PFBC with impaired cellular Pi homeostasis, possibly via SLC20A2.*')